**communications**

**biology**

# An intrinsically disordered nascent protein interacts with specific regions of the ribosomal surface near the exit tunnel

Valeria Guzman-Luna [1], Andrew M. Fuchs[1], Anna J. Allen[1], Alexios Staikos[1] & Silvia Cavagnero [1✉]

The influence of the ribosome on nascent chains is poorly understood, especially in the case of proteins devoid of signal or arrest sequences. Here, we provide explicit evidence for the interaction of specific ribosomal proteins with ribosome-bound nascent chains (RNCs). We target RNCs pertaining to the intrinsically disordered protein PIR and a number of mutants bearing a variable net charge. All the constructs analyzed in this work lack N-terminal signal sequences. By a combination chemical crosslinking and Western-blotting, we find that all RNCs interact with ribosomal protein L23 and that longer nascent chains also weakly interact with L29. The interacting proteins are spatially clustered on a specific region of the large ribosomal subunit, close to the exit tunnel. Based on chain-length-dependence and mutational studies, we find that the interactions with L23 persist despite drastic variations in RNC sequence. Importantly, we also find that the interactions are highly $Mg^{+2}$-concentration-dependent. This work is significant because it unravels a novel role of the ribosome, which is shown to engage with the nascent protein chain even in the absence of signal or arrest sequences.

[1] Department of Chemistry, University of Wisconsin-Madison, 1101 University Ave., Madison, WI 53706, USA. ✉email: cavagnero@chem.wisc.edu

The mechanism of in vitro protein refolding, starting from chemically or thermally denatured states, has been studied at length and is reasonably well understood to date[1]. On the other hand, we still know very little about how proteins fold within the cellular environment. For instance, we do not yet have a clear view of how ribosome-bound nascent chains (RNCs) emerge out of the ribosomal exit tunnel (Fig. 1a) and whether this process involves interactions with the ribosomal surface. Recent investigations report the influence of the ribosome on the earliest stages of protein folding[2–4]. For instance, the ribosome contributes to decreasing cotranslational protein aggregation, enabling higher yields of folded protein[5]. Further, the translation machinery as a whole plays a role, given that cotranslational

protein folding rates can be modulated by fast- and slow-translating codons[6–9], and synonymous codon substitutions might challenge the cellular chaperone machinery[10]. In all, however, the structural aspects and requirements for co-translational protein folding are far from being fully understood.

The role of interactions between RNCs and ribosomal proteins is of special interest as these interactions may influence the rate of translation as well as nascent-protein characteristics. Some of these interactions have been detected so far. For instance, contacts between ribosomal proteins and the 24-residue TnaC ribosomal-arrest peptide were identified via crosslinking and single-particle cryo-electron microscopy[11,12]. The TnaC sequence stalls translating ribosomes and establishes strong noncovalent contacts with

**Fig. 1 Prominent features of ribosome, ribosome-bound nascent chains (RNCs) and EDC-mediated chemical crosslinking. a** Crystal structure of the *E. coli* ribosome (PDB ID: 4YBB). Ribosomal RNA (23S and 5S RNA: teal; 16S RNA: light green) and ribosomal proteins (50S rProteins: purple; 30S rProteins: bright lime green) are color-coded to facilitate identification. **b** Cartoon illustrating the conformationally dynamic and static IDP populations of ribosome-bound PIR nascent chains identified via fluorescence depolarization decay in the frequency domain[22]. **c** Overview of the EDC-mediated crosslinking process, enabling the nonspecific crosslinking of carboxylates to primary amines within a 1 to 5 Å distance. Species are represented at pH 7.0. **d** Schematic illustration of key experimental steps involving resuspended PIR RNC analysis via low-pH SDS-PAGE followed by selective detection of PIR RNCs by fluorescence emission. Lanes 1, 2, 3, and 4 display untreated RNCs, RNCs released from the ribosome via the antibiotic puromycin, outcome of crosslinking including RNC-ribosomal protein crosslinked adducts, and an experimental control, respectively.

ribosomal proteins L4 and L22 within the ribosome exit tunnel[11–13]. Further, the 17-residue *E. coli* ribosomal stalling sequence of the SecM protein interacts with both L4 and L22 at the ribosomal-tunnel constriction site, and again with L22 in proximity of the exit-tunnel vestibule[14].

Explicit interactions inside and outside the ribosomal exit tunnel were also observed for nascent proteins bearing signal sequences at the N terminus[15]. This class of proteins accounts for a third of the *E. coli* proteome[15]. The signal sequence assists the targeting of proteins towards membrane insertion or secretion to the trans side of the plasma membrane, mostly via the Sec pathway[16]. Houben et al. showed that constructs derived from the Lep leader peptidase protein (ranging from 9 to 50 residues) and bearing at least 50% of the N-terminal signal sequence interact and covalently crosslink to the L4 and L22 ribosomal proteins within the exit tunnel and then to L23 outside the exit tunnel, as the nascent protein elongates[17]. Additional studies also showed that nascent polypeptides bearing N-terminal signal or translation-stalling sequences experience interactions with the L23, L24, and/or L29 ribosomal proteins outside[18–20] — and in some cases inside[18,20] — the ribosomal exit tunnel. Some of the above interactions are known to facilitate nascent-chain stalling or mediate downstream conformational changes on the nascent chain[19]. Table S1 provides a summary of the known ribosomal proteins interacting with RNCs that bear translational-stalling or signal sequences. Given that the Sec-pathway-dependent proteins, which are not cytosolic, are exported in an unfolded state[16], it is possible that the aforementioned interactions are different, in the case of cytosolic nascent proteins.

Fluorescence anisotropy decay studies were carried out on RNCs of the intrinsically disordered protein (IDP) PIR, i.e., the N-terminal domain of the phosphorylated insulin receptor, which interacts only weakly with cotranslationally active chaperones. The amplitude of the local motions of the PIR nascent-chain N terminus is highly spatially confined[21]. More detailed experiments showed that two populations of widely different local mobility (91% static and 9% dynamic)[22] are present, as schematically illustrated in Fig. 1b. This result suggests the presence of ribosome-interacting (static) and non-interacting (dynamic) species in solution.

NMR linebroadening and transverse-relaxatio-rate investigations on selective nascent-protein residues and molecular-dynamics simulations led to propose that RNCs of immunoglobulin-like domains and α-synuclein bearing the SecM translational-arrest motif have N-terminal regions that experience transient interactions with the *E. coli* ribosomal surface close to the exit tunnel[23–26]. Further, RNCs of T4 lysozyme bound to the ribosome via a C-terminal link exhibit slower folding rates than the ribosome-released full-length protein lacking the C-terminal link[27]. This result, in turn, led the authors to suggest interactions with the ribosomal surface[27].

Despite the significance of the above studies, there is currently a crucial lack of information and no explicit evidence about noncovalent contacts between nascent proteins lacking signal or arrest sequences and the ribosomal surface.

Here, we take initial steps to address this lack of knowledge by probing contacts between RNCs derived from PIR, an IDP, and ribosomal proteins. PIR is a 91-residue IDP that lacks a well-defined secondary and tertiary structure in the absence of its binding partner and was reported not to cotranslationally interact with molecular chaperones at low chaperone concentration[21,28,29]. Hence it is an ideal model system for exploring interactions with ribosomal proteins devoid of confounding variables, e.g., co-translational protein compaction/folding or binding to molecular chaperones. By a combination of fluorescence-detected chemical crosslinking and western blotting,

we identified explicit interactions between the PIR nascent chain and the L23 and L29 ribosomal proteins. The interactions are chain-length-dependent, given that only longer chains interact with L29, and are clustered within a distinct region of the large ribosomal subunit. RNC-net-charge and $Mg^{+2}$ concentration-dependence studies concurred in showing that the interactions between PIR RNCs and the ribosome are largely $Mg^{+2}$-mediated and are likely to involve negatively charged nascent-chain residues. The discovery of interactions between nascent proteins lacking signal or arrest sequences and the ribosome, together with their $Mg^{+2}$ dependence, contribute to advance our knowledge on fundamental aspects of the cellular milieu.

## Results

**Experimental design**. Ribosomal nascent chains of the intrinsically disordered protein PIR (PIR RNCs) were generated in an *E. coli* cell-free system via oligonucleotide-assisted mRNA cleavage[30,31] (see Methods section). RNCs were specifically labeled at the N terminus with the BODIPY-502 fluorophore, to enable SDS-PAGE detection via fluorescence imaging. Previous work based on fluorescence anisotropy decay provided the inspiration for the present study by identifying conformationally biased PIR RNCs (static population, 91%, Fig. 1b)[21,22]. These studies suggested the presence of noncovalent intra- or inter-molecular constraints.

To directly probe for the presence of any interactions between RNCs and the ribosome, we used the zero-length crosslinker carbodiimide 1-ethyl-3-[3-dimethylaminopropyl] carbodiimide hydrochloride (EDC). EDC is a water-soluble small molecule that serves as a reporter of intra- and inter-molecular interactions in proteins[32,33]. As shown in Fig. 1c, EDC covalently couples carboxylic acids to primary amines that are spatially separated by 1−5 Å, and serves as a sensor for neighboring noncovalent interactions by establishing an irreversible covalent link in the vicinity of the interaction site. Briefly, EDC reacts rapidly with carboxylic functional groups of D and E residues (or the C terminal) to yield an *O*-acylisourea intermediate. The latter can either hydrolyze to regenerate the starting materials (Fig. 1c, left branch), or couple to a nearby primary amine belonging to the K side-chain (or to the primary amine of the protein N terminal) to generate an effectively irreversible[34] amide bond (Fig. 1c, right branch).

All experiments were carried out on purified ribosome-stalled RNCs[30] and were analyzed with low-pH SDS-PAGE to prevent fluorophore hydrolysis[35]. The ribosome-bound nature of RNCs was assessed upon treatment with puromycin as described[36]. This small antibiotic covalently links to the nascent-chain C terminus[37] when the A and P sites of the 50S subunit have an intact structure[38], leading to ribosome release and to a dramatic drop in molecular mass due to tRNA loss (compare gel lanes 1 and 2 of Fig. 1d).

A strength of our strategy is the ability to sample all protein–protein interactions in solution, given that EDC diffuses freely and can capture noncovalent contacts involving polypeptides and proteins (including RNCs, ribosomal proteins, and chaperones). At the same time, the N-terminal RNC-fluorophore labeling enables the selective visualization of crosslinked proteins that experience interactions with RNCs, greatly simplifying the overall analysis.

**Nascent protein chains interact with ribosomal proteins**. When resuspended PIR RNCs, generated in a WT *E.. coli* cell-free system, were treated with the EDC crosslinking agent, a new gel band of higher molecular mass was detected (Fig. 1d, compare lanes 1 and 3). This band results from EDC-mediated

crosslinking and is diagnostic of interactions between nascent PIR and one or more protein. EDC is insensitive to interactions with ribosomal RNA (rRNA), given that (a) imidazole is required to establish crosslinks between the RNA 5' phosphate and aliphatic amines within proteins[32], and (b) our samples contained no imidazole. Interactions of PIR RNCs with rRNA will be addressed elsewhere.

A comparison between lanes 2 and 4 of Fig. 1d shows that ribosome-released PIR (lane 2) undergoes no crosslinking when EDC treatment occurs after puromycin-induced ribosome release (lane 4). Therefore, we conclude that nascent PIR needs to be ribosome-associated, for crosslinking to occur (Fig. 1d, lane 3). Intramolecular crosslinking of ribosome-released PIR is ruled out because ribosome-released PIR has identical gel-migration patterns regardless of whether EDC is added, after ribosome-release (lanes 2 and 4).

In general, three main independent events contribute to the probability of crosslinking ($P_{xl}$), i.e., the intrinsic reactivity of the D, E, and K residues, expressed as reaction probability $P_{reac}$, the probability of protein exposure to solvent and other solution components ($P_{exp}$), which describes the probability of crosslinker access to reactive functional groups, and the probability that an actual interaction in proximity of the crosslinking site exists ($P_{int}$). This scenario is consistent with the expression

$$P_{xl} = P_{reac} \: x \: P_{exp} \: x \: P_{int}, \qquad (1)$$

The D, E, and K amino-acid side chains have similar intrinsic reactivity[33], hence similar $P_{reac}$. Therefore, observed differences in crosslinking outcomes are solely due to a combination of differences in $P_{exp}$ and $P_{int}$. Consequently, due to variable contributions of $P_{exp}$, it is expected that crosslinking only provides a qualitative assessment for the presence of interactions. The relative interacting population ratios, therefore, may not be accurately quantified.

After cell-free transcription-translation, our PIR$_{91}$ RNCs co-pellet with negligible amounts of trigger factor (TF) and Hsp70 (i.e., DnaK in *E. coli*) chaperones. Concentrations of TF and Hsp70 in resuspended PIR$_{91}$ RNCs range between 2 and 15 nM[31]. Under these conditions, crosslinking with the cotranslationally active Hsp70 (70 kDa) and/or TF (48 kDa) is experimentally ruled out based on the fact that RNCs from cell strains including or lacking the TF chaperone have identical crosslinking patterns (Fig. S1a) and lack any band at the approximate expected higher molecular weight. Experiments conducted in the presence of larger added concentrations of TF (up to 40 μM) or DnaK/DnaJ/GrpE (up to 50/10/25 μM) chaperones revealed the presence of weak RNC-TF and RNC-DnaK interactions. The apparent dissociation constants pertinent to these interactions are $K_{d,app}$ = 3.8 ± 0.6 μM and 109 ± 22 μM for TF and K/J/E, respectively (Fig. S1b). These values are at least 1 or 2 orders of magnitude larger than typical $K_d$ values previously detected for the interaction of either TF ($K_d = 0.053-0.69$ μM)[39] or ADP-DnaK ($K_d = 0.1-1.0$ μM)[40] with a variety of other proteins in physiologically relevant environments. Hence, due to the expected stronger interactions with other proteins in the cellular milieu, the cellular TF, and DnaK concentration bioavailable (i.e., non-client-protein bound) for interaction with PIR$_{91}$ is expected to be significantly lower than the maximum chaperone concentrations tested here (Fig. S1b, up to 40–50 μM). In summary, some weak interactions of PIR$_{91}$ RNCs with TF and DnaK chaperones are likely present in crude cell-free systems. However, given the moderate extent of these interactions and the likely low population of TF-RNC and DnaK-RNC complexes under cell-relevant conditions, we elected to focus primarily on contacts between PIR RNCs and ribosomal proteins, in this work.

Importantly, 29 out of 55 ribosomal proteins have an 11–21 KDa size compatible with the observed increase in molecular weight of the crosslinked band (Table S2)[13]. Hence, we conclude that the RNC interacting partners detected in lane 3 of the gel in Fig. 1d are likely ribosomal proteins, either individual or in combination[41]. Finally, the interactions identified here are not induced by the BODIPY-502 fluorophore, which is covalently linked to the RNC N terminus, as it is known that this fluorophore does not interact with the ribosome[21].

**The kinetics of EDC incorporation follows a crosslinking mechanism involving a burst phase.** The progress of the crosslinking process was followed over time (Fig. 2a). Interestingly, ~60% of the crosslinked RNCs are generated fast, within a burst phase that accounts for the manual-mixing dead time of ca. 0.1 min. The burst phase is followed by a slow detectable phase of more moderate amplitude. The raw data, illustrating the fraction of crosslinking ($F_{XL}$) vs time (Fig. 2a), were fit to the simplest kinetic model compatible with fluorescence anisotropy decay data[22] and with key aspects of the known EDC crosslinking mechanism (Fig. 1c)[33].

Based on the established fast appearance of fully crosslinked species at 0.1 s (see relevant gel band in Fig. 2a), the burst-phase events must include activation of solvent-exposed carboxylates, known to be fast ($3.3 \times 10^3$ M$^{-1}$ s$^{-1}$)[33], followed by a second crosslinking step establishing the covalent link. Given the efficient formation of the initial crosslinked population $F_{fast}$, we assumed negligible or very fast nascent-chain or ribosomal conformational changes to take place during this time. The burst-phase events were of course not subject to curve fitting, given the inability to explicitly follow their timecourse.

The slow phase (black reactions in Fig. 2b) was modeled as two steps. The first step, characterized by rate constant $k_1$, accounts for a putative conformational change, and is ultimately responsible for the slow formation of the $F_{slow}$ crosslinked population. The second step, which accounts for the necessary final stage of crosslinking, was assigned a fixed and large rate-constant value ($k_2$, see Methods section). Note that this step must also occur rapidly within the burst phase (Fig. 2b). Within the detectable phase, the step with rate constant $k_2$ is much faster than the preceding step (with rate constant $k_1$), hence kinetically inconsequential. The fairly long lifetime $\tau_1$ of 255 s (=4.25 min) of the first step, derived from curve fitting, justifies the conclusion that this step dominates the observable kinetics. The slow conformational change may arise from suboptimal shape, due to unoptimized carboxylate-activated-location(s), of nascent-chain species D (Fig. 2b) towards the second crosslinking step. The residuals and reduced chi-square values for the fits, justifying the choice of the simple model in Fig. 2b, are displayed in Fig. S2d, e, respectively.

The crosslinked fractions at 30 min, in combination with the static and dynamic equilibrium populations assessed by fluorescence anisotropy decay in the frequency domain (Fig. 2c), show that the interacting populations are unequivocally identified only at a qualitative level. Hence, this method is generally appropriate to detect the presence of interactions, but it is unable to provide a quantitative assessment of the populations of interacting and non-interacting species. On the other hand, the presence of subpopulations $F_{fast}$ and $F_{slow}$ within the interacting nascent chains, detected by crosslinking, suggests the presence of conformational heterogeneity within the bound subpopulation. This feature of the nascent-chain ensemble was undetectable by fluorescence anisotropy.

Finally, we performed kinetic simulations based on the parameters adopted in the model of Fig. 2b and the value of $k_1$

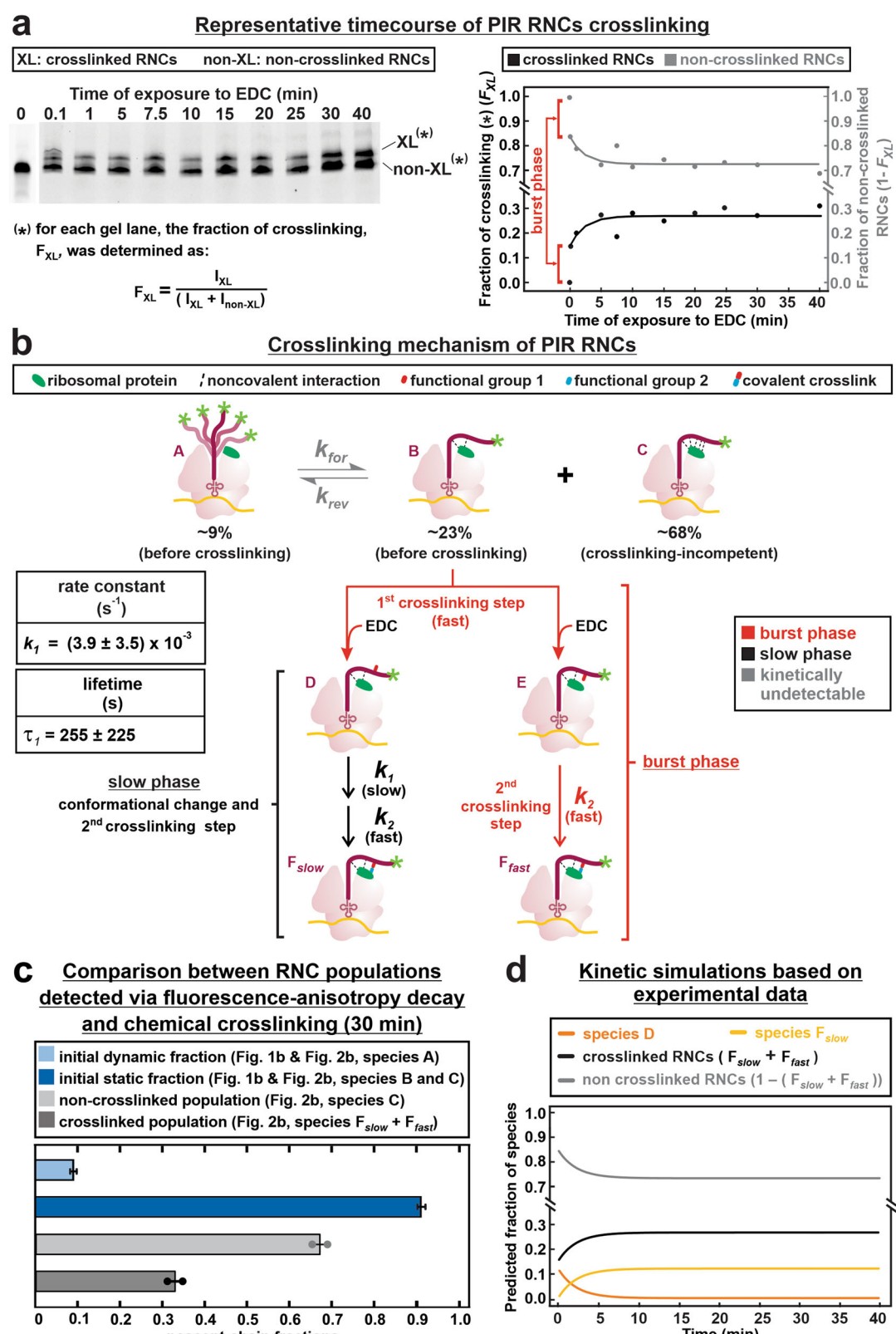

**Fig. 2 Timecourse and proposed mechanism of RNC crosslinking. a** SDS-PAGE analysis and corresponding plot illustrating the timecourse of EDC-mediated PIR RNC crosslinking. See Methods section for details. **b** Proposed mechanism of PIR RNC crosslinking and rate constant ($k_1$; avg ± S.E., for $n = 2$) with corresponding lifetime ($\tau_1$) value derived from data fitting. Gray, orange, and black arrows denote steps that were not explicitly modeled, burst-phase steps, and explicitly-modeled slow steps, respectively. **c** Fractions of conformationally dynamic and static RNC populations assessed via fluorescence anisotropy decay[22] (light and dark blue, respectively), and fractions of non-crosslinked and crosslinked PIR RNCs upon 30 min (light and dark gray, respectively) exposure to EDC. Data are displayed as avg ± SE ($n = 2$). **d** Kinetic simulations employing, as input, the rate constants derived from fitting of the data in panel **a**. See Fig. S2a–c for additional simulations.

**Fig. 3 Identification of specific ribosomal proteins crosslinked to nascent PIR$_{91}$ via SDS-PAGE and western blotting.** Each lane displays low-pH 9% SDS-PAGE analysis of N-terminal fluorescently-labeled PIR RNCs generated via transcription-translation in an *E. coli* WT S30 cell-free system (left) followed by western blotting (right) employing antibodies against ribosomal proteins **a** L17, **b** L18/L22, **c** L23, **d** L24, **e** L29, and **f** L32. See Methods section for additional details. Representative data, out of $n = 2$, are displayed.

derived from curve fitting. As shown in Fig. 2d (black and gray traces, see also Fig. S2c), the simulations closely match the experimental data. In addition, they show the timecourse of disappearance of the slowly-crosslinking species D as well as the appearance of the fully crosslinked $F_{slow}$ population.

In summary, the kinetic data and the accompanying model highlight the fact that a considerable fraction of the RNC population crosslinks very fast while another smaller fraction does so more slowly. Importantly, the model acknowledges the experimental observation that some of the RNC population never crosslinks over the timescale of our experiments, as evident in the XL and non-XL gel bands in Fig. 2a at the longest crosslinking time of 40 min.

**PIR RNCs interact with ribosomal proteins L23 and L29, and the interaction region is clustered within a specific region of the 50S ribosomal subunit.** Next, we focused on investigating which ribosomal proteins interact with PIR RNCs. Figure 3 shows the results of crosslinking experiments analyzed via SDS-PAGE (left) and western blotting (right). Antibodies against ribosomal proteins within the *E. coli* 50S ribosomal subunit were employed in western blots. SDS-PAGE shows that the apparent molecular weight of the crosslinked peptidyl-tRNAs (~44 kDa, lanes 3) decreases significantly (to ~27 kDa, lanes 4) after treatment with puromycin, a small-molecule antibiotic that eliminates the tRNA

component of peptidyl tRNAs within the ribosome. Taking into account the molecular weight of full-length PIR (~9 kDa) and the thickness of the SDS-PAGE gel bands in lanes 4, the total molecular weight of the crosslinked proteins is estimated to vary between 11 and 23 kDa. This value should be regarded as approximate, given that the crosslinked complex may display somewhat altered migration through the gel.

The identity of ribosomal proteins crosslinked to nascent chains was assessed upon comparing SDS-PAGE lanes 3 and 4 to the corresponding western blots. A control exposing RNCs to EDC only after puromycin confirms that the observed crosslinking is a peculiar feature of ribosome-bound nascent proteins, given that ribosome-released RNCs do not undergo any cross-linking (compare SDS-PAGE gels lanes 2 and 5).

The western blots of Fig. 3 reveal that full-length ribosome-bound nascent PIR (PIR$_{91}$) crosslinks to ribosomal proteins L23 and L29 (see arrows in panels c and e) but not to L17, L18/L22, L24, and L32 (see arrows in panels a, b, d, and f). The western blot band supporting RNC crosslinking to L23 exhibits a considerably higher relative intensity than in the case of L29, suggesting that PIR$_{91}$ interacts strongly with L23 but very weakly with L29.

A mapping of all ribosomal proteins on the face of the 50S subunit hosting the tunnel vestibule and tunnel exit is shown in panels a and d of Fig. 4. A color-coded representation of all the nonpolar and EDC-reactive residues of the 50S ribosomal subunit (Fig. 4b) shows that, for most proteins, the EDC-reactive charged

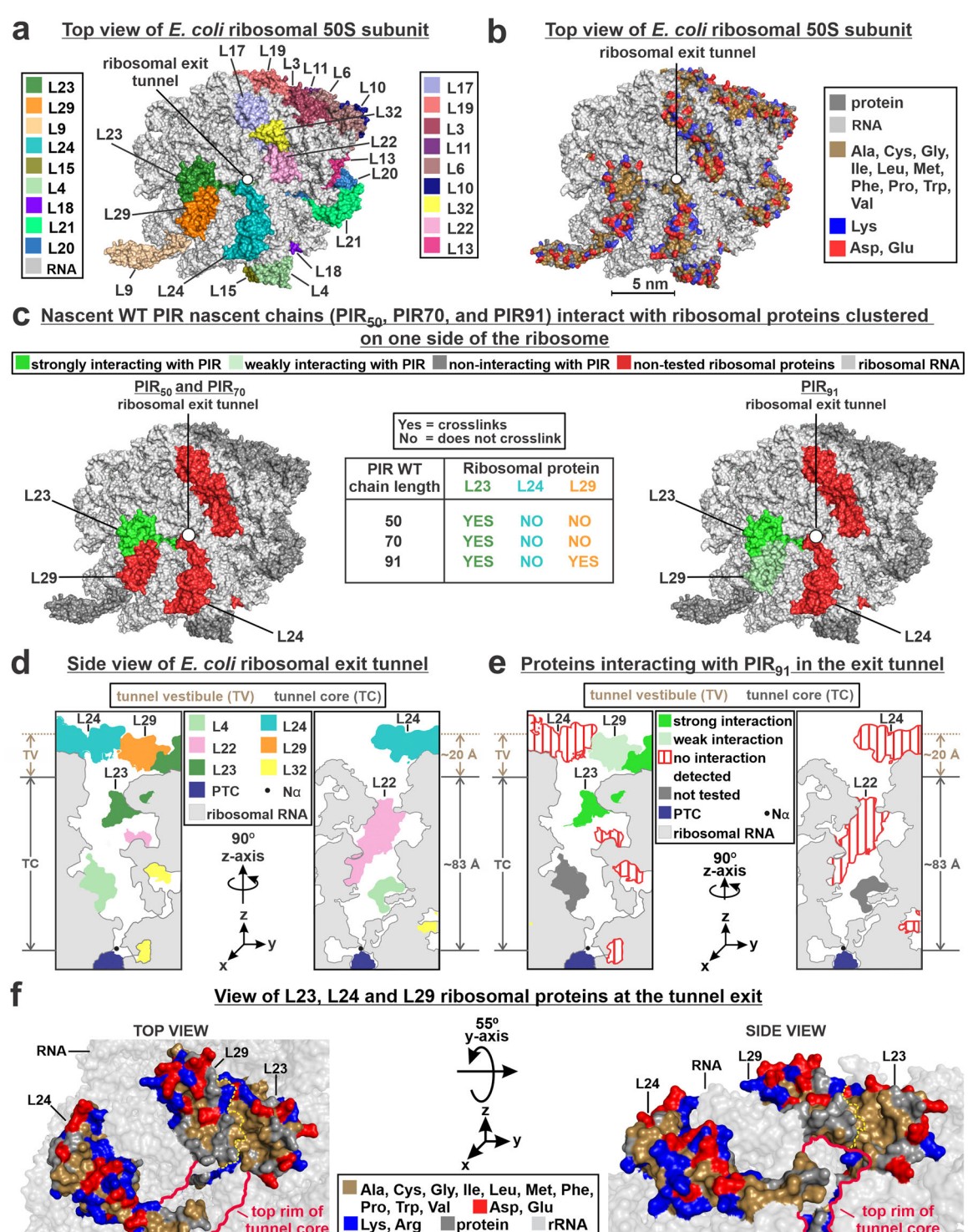

**Fig. 4 Structural representation of ribosomal proteins interacting with PIR RNCs. a** Top view of crystal structure of the *E. coli* 50S ribosomal subunit (PDB ID: 4YBB) including color coding of ribosomal-proteins. The ribosomal-tunnel exit is denoted by a white circle. **b** crystal structure of the *E. coli* 50S ribosomal subunit (PDB ID: 4YBB) highlighting the distribution of nonpolar amino acids and EDC-reactive residues (Blue: Lys, Red: Asp, Glu) for the ribosomal proteins surrounding the exit tunnel. **c** Top view of *E. coli* 50S ribosomal subunit (PDB ID: 4YBB) highlighting ribosomal proteins interacting strongly (dark green), weakly (light green) and non-interacting (red) with PIR$_{50}$, PIR$_{70}$, and PIR$_{91}$ PIR RNCs. Also shown is a Table summarizing the chain-length dependence of the interactions between PIR RNCs and ribosomal proteins assessed via SDS-PAGE and western blotting (see Fig. S3). **d** Side view of *E. coli* ribosomal exit tunnel core[53,75,94] (PDB ID: 4YBB) and tunnel vestibule highlighting ribosomal proteins L4, L22, L23, L24, L29, and L32. The peptidyl transferase center (PTC) is shown in blue and the N$^{\alpha}$ of the C-terminal residue of a growing polypeptide chain is denoted by a black dot. **e** Side view of *E. coli* ribosomal exit-tunnel core[53,75,94] (PDB ID: 4YBB) and tunnel vestibule highlighting ribosomal proteins interacting strongly, weakly, and non-interacting with PIR RNCs. **f** Close-up view of L23, L24, and L29 ribosomal proteins, which are located within the *E. coli* 50S ribosomal exit tunnel (PDB ID: 4YBB). See Movie S1 for a more informative interactive view.

residues are fairly uniformly distributed. This result strongly suggests that the crosslinking between RNCs and ribosomal proteins L23 and L29 (Fig. 4c, image on righthandside) is not biased by differences in EDC reactivity. Close inspection of the amino acid character of proteins L23 and L29 (Fig. 4b) reveals the presence of a solvent-exposed nonpolar patch that includes a significant portion of L23 and a smaller part of L29. Interestingly, this hydrophobic region is located at the interface between the tunnel core and the tunnel vestibule, as shown in Fig. 4f (see also Movie S1). It is worth noting that proteins L22 and L24 were found not to crosslink to nascent PIR, despite being large contributors to the tunnel-core and tunnel-vestibule surface areas, respectively (Fig. 4d, e)[42]. As discussed later, the lack of detected interactions with L22 may arise from poor accessibility of the tunnel-core region.

A major outcome of this analysis is that the ribosomal proteins interacting with $PIR_{91}$ RNCs are clustered within selected regions of the outer ribosomal surface and exit tunnel (Fig. 4c–e).

**Shorter, incomplete-length PIR RNCs interact only with ribosomal protein L23.** To assess the chain-length dependence of the RNC interactions with ribosomal proteins, we carried out crosslinking and western-blot experiments on non-full-length PIR RNCs bearing 50 and 70 residues (Fig. S3), and compared the results with full-length PIR RNCs. Nascent $PIR_{70}$ and $PIR_{50}$ crosslinked exclusively with L23 (lefthandside image of Figs. 4c and S3). In addition, no interactions with ribosomal proteins L24 or L29, which are clustered on the same side of the ribosomal surface, were detected (Fig. S3). The Table of Fig. 4c shows that all the tested chain lengths interact with L23 while only full-length PIR ($PIR_{91}$) interacts weakly with L29. Hence, even short RNCs lacking signal or arrest sequences can interact with the ribosome. It is worth noting that RNCs as short as 50 residues are expected to have a portion of their chain accessible to proteolytic cleavage[43], hence likely also available for crosslinking-detectable interactions within the accessible regions of tunnel vestibule and(or) outer ribosomal surface.

**The extent of PIR RNC interactions with the ribosomal surface increases with chain length.** To more thoroughly characterize the global nascent-chain dependence of the interactions between RNCs and ribosomal proteins, several additional PIR RNCs of variable length (from 30 to 91 residues) were subject to cross-linking. Representative raw data are shown in Fig. 5a. A quantitative outlook of the crosslinking fractions of PIR RNCs as a function of chain length is shown in Fig. 5b (see also Fig. S4). The plot on the righthandside of Fig. 5b and the accompanying $t$-test (Fig. S5) are particularly informative as they show that the extent of detected interactions, when normalized so that they are reported on a per-EDC-reactive residue basis, display a statistically significant increase as RNC chains get longer. Given that the reactivity of RNCs and ribosomal proteins is roughly uniformly distributed (Fig. 4b and S7), the above result suggests that longer nascent chains have a stronger affinity per residue for the ribosomal surface, as expected based on simple multivalency arguments[44].

The 30- and 40-residue RNCs are known to be buried[43] within the narrow (10.3−16 Å), ribosomal tunnel core[45,46]. Hence, these short nascent chains are expected to experience less efficient crosslinking due to decreased accessibility to the relatively large EDC crosslinker (diameter ca. 8.7 Å). Poor accessibility is likely further exacerbated by the presence of the nascent chain. The expectation of poor crosslinking efficiency of shorter (30–40 amino acids) RNCs is consistent with the computationally predicted slow diffusion of solvent across the tunnel core[47]

and with previous experimental accessibility assays on other RNCs[48–50].

**Full-length PIR RNCs interact mainly with the solvent exposed portion of the ribosome.** Figure 5c shows a schematic representation of an extended-chain RNC emerging out of the ribosomal tunnel, considering that each amino acid contributes 3.63 Å to the overall length[51]. The sphere represents the 1–2 N-terminal amino acids bound to the BODIPY-502 fluorophore. The cone semi-angle parameter $\theta$, which describes the amplitude of the local sub-ns motions of the N terminus within the ribosome-bound state (sphere in Fig. 5c), was previously assessed experimentally by fluorescence anisotropy decay in the frequency domain[21]. The results of this analysis are reproduced in Fig. 5d. Briefly, the plot shows two distinct regions, and the pronounced variation in the slope of the graph at 29 ± 3 residues shows that RNCs of length 37 through 91 experience a distinctly different local environment. This environment is characterized by considerably wider-amplitude N-terminal local motions than shorter chains, strongly suggesting that PIR RNCs bearing 37–91 residues are no longer located in the tunnel core. See Fig. S6 for a complete analysis.

Further insights are provided by plots of the crosslinking fraction as a function of number of PIR EDC-reactive residues in each region of the ribosome (Fig. 5e). Remarkably, when a fully extended RNC chain is assumed, the fraction of crosslinking correlates linearly ($R^2 = 0.88$) with the number of PIR EDC-reactive residues across the L23 and L29 protein surface outside the tunnel (SOT region, Figs. 5e and S7a-d). A comparably good correlation ($R^2 = 0.80$) is also obtained when a compact PIR chain is assumed to interact with the L23 and L29 protein surface in the tunnel vestibule (TV region, Fig. 5e).

In all, the data in Fig. 5a–e are consistent with either a model where full-length PIR RNCs interact with ribosomal-protein regions on the outer surface of the ribosome (Model I in Fig. 5f), or a model according to which compact PIR RNCs interact preferentially with the tunnel vestibule (Model II in Fig. 5f). Upon considering (i) the ribosomal-protein location mapped in Fig. 4 (panels c–e) and (ii) the fact that full-length PIR chains interact with both L23 and L29 while $PIR_{50}$ and $PIR_{70}$ only interact with L23, it follows that both Models I and II (or a combination thereof) describe viable interaction settings. Additional scenarios including a more structured backbone within the tunnel core, e.g., Models III and IV, are also formally feasible. These models are consistent with previous reports showing some helical content in RNCs bearing foldable protein sequences[48,52–58]. However, in the case of the intrinsically disordered PIR RNC, models III and IV are inconsistent with the 29 ± 3 residues slope change of the cone semi-angle data of Fig. 5d. In addition, Models III or IV, which assume ca. 68 helical residues in the tunnel core, seem unfit to accommodate the completely isotropic motions that were previously reported for the dynamic conformation of full-length PIR RNCs[22]. Therefore, Models III and IV are regarded as unlikely, though they cannot be formally ruled out.

In summary, Models I and II are the most plausible. For simplicity, the data discussed next will assume Model I.

**Even highly negatively charged PIR RNCs interact extensively with the ribosomal surface.** Next, we explored the dependence of the detected interactions between PIR RNCs and ribosomal proteins on RNC net charge. Due to the known phenomenon of charge segregation within the ribosome, (Fig. S8 and Movie S2), negatively charged carboxyl groups of ribosomal proteins point towards the solvent surface[59]. Conversely, most positively charged amino groups of ribosomal proteins face the negatively

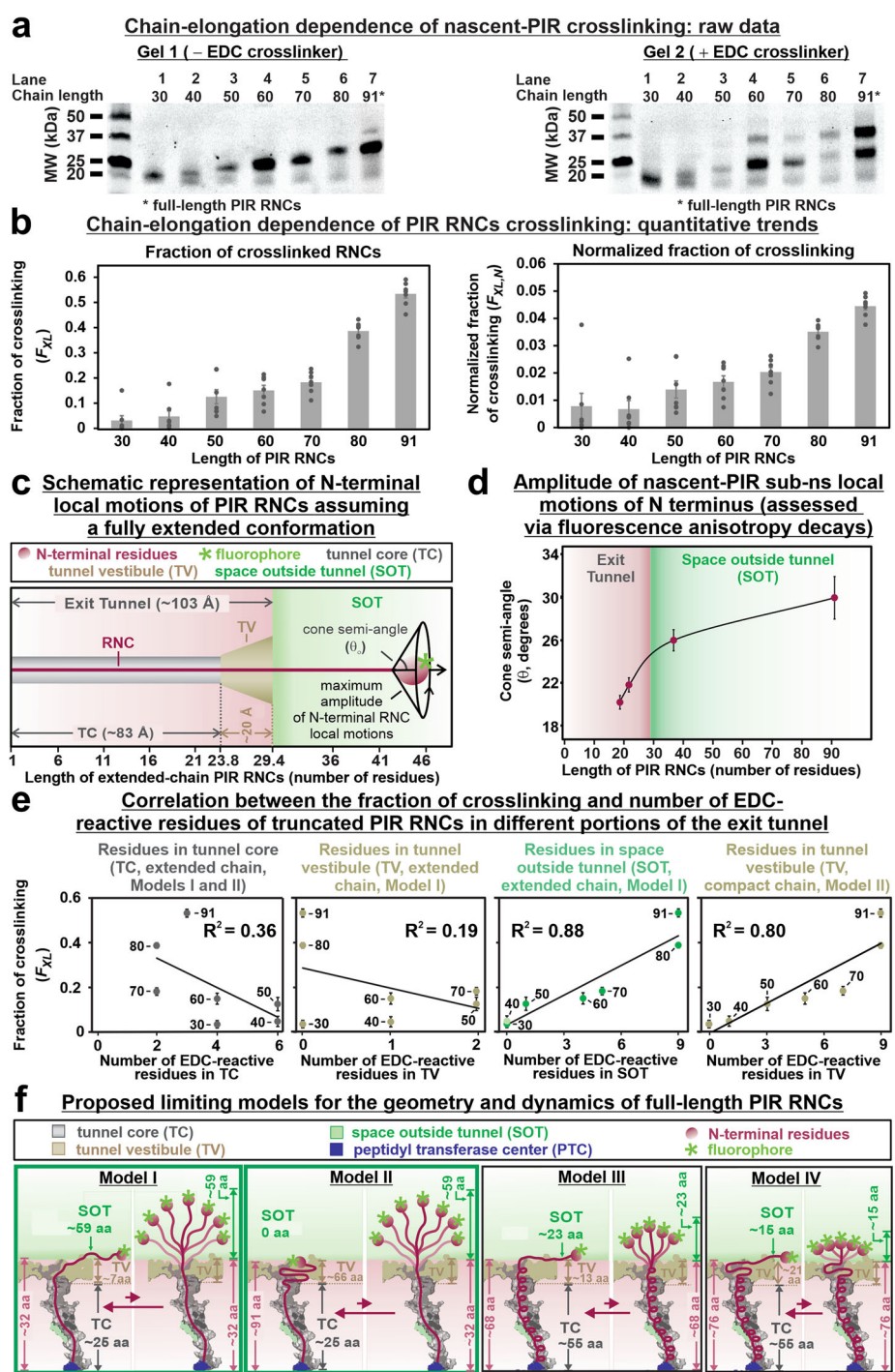

**Fig. 5 Chain-length dependence of fraction of crosslinked PIR RNCs and proposed models taking into account data on ribosomal-complex geometry and dynamics. a** SDS-PAGE of the PIR RNCs of 30–91 chain length in absence and presence of the EDC crosslinker. **b** Crosslinking fraction of PIR truncated chains (left) calculated as in Fig. 2a. Error bars denote ± SE for $n = 7$. Normalized crosslinking fractions of PIR RNCs chains of 30–91 aa (right), calculated by dividing the crosslinking fraction by the total number of EDC-reactive residues per each RNC. The p values for the two-tailed $t$-test ($P < 0.05$) are in Fig. S5. **c** Representation of a transverse section of the ribosomal exit tunnel showing consensus dimensions[53] and regions of the Tunnel Core (TC), the Tunnel Vestibule (TV), and the Space Outside of Tunnel (SOT). A fully extended PIR RNC (3.63 Å per residue)[51] shows the amplitude of N-terminal nascent-chain motions described by the cone semi-angle determined via time-resolved fluorescence depolarization[22]. **d** Previously determined cone semi-angles values describing the amplitude of nascent-chain N-terminal motions (sub-ns) of the dominant conformation, i.e., the biased species[21,22]. The range of residues corresponding to the different tunnel regions are also shown (see Fig. S6 for details). **e** Assessment of correlation between fraction of crosslinking and number of EDC-reactive residues of PIR RNCs in different portions of the exit tunnel. **f** Limiting models for the nascent-PIR$_{91}$ conformational ensemble. The exit tunnel is shaped according to the crystal structure of the *E. coli* 50S ribosome subunit (PDB ID: 4YBB). The peptidyl transferase center (PTC) is shown in dark blue and the highly conserved tunnel constriction formed by the L4 and L22 ribosomal proteins are displayed in dark-gray and green, respectively[95].

charged rRNA core[59]. Hence, the outer surface of the ribosome is highly negatively charged due to both ribosomal proteins and ribosomal RNA (Figs. 6 and S9).

We analyzed a series of RNC variants with total net charge (at pH 7.4) ranging progressively from +1.1, as in $PIR_{91}$ 0.0.0 (wild type, WT), to −10.9, as in the most negatively charged variant, denoted as $PIR_{91}$ 1.2.3. The sequence of the different constructs is shown in Fig. 6a. The same variants were previously found to exhibit a progressively lower fraction of the conformationally biased component (Fig. 1b) for more negatively charged RNCs, via fluorescence anisotropy decay in the frequency domain[22]. Here, we focused on a qualitative evaluation of the interacting species upon introduction of increasing numbers of negative charges (Fig. 6a).

No significant variations in crosslinking fractions were observed ($P < 0.05$, two-tailed $t$-test) for wild-type and variant PIR RNCs bearing net charges ranging from +1 to −6.9 ($PIR_{91}$ 0.0.0–1.0.3, Fig. 6a–c), and a fairly modest decrease in interaction was detected upon modifying the RNC net charge from −6.9 to 10.9. This remarkable lack of response to dramatic variations in RNC net charge suggests that classical electrostatic interactions, including ion pairs and negative-to-negative charge repulsions, between ribosome and RNCs do not play a dominant role. The persistence of significant interactions with the ribosome even for the highly negatively charged PIR 1.2.3 RNC is particularly surprising, given the highly negatively charged ribosomal surface.

Previous ionic-strength-dependence studies[22] and additional monovalent salt-dependence investigations at variable KCl concentration carried out here (Fig. 7a) show an overall lack of dependence on overall ionic strength. Note that the tested $K^+$ levels (500 mM) were even higher than the known $E.$ $coli$ $K^+$ cytoplasmic concentration (180–200 mM)[60]. The above results can be rationalized in light of the thick layer of counterions that is known to be present around RNA and RNA-protein complexes[61]. These counterions are more highly concentrated near RNA-rich macromolecular surfaces than in the bulk, and effectively shield electrostatic interactions[62–64]. This effect explains the weak ionic-strength dependence observed in Fig. 7a, given that a concentrated counterion layer is likely present even in the absence of added KCl.

Figure 6e shows two versions of the ribosomal electrostatic surface potential bearing different scales ($±5$ $k_BT/e$ and $±10$ $k_BT/e$), where $k_B$ is the Boltzmann constant and $e$ is the electric charge of the electron. These images show that the major interacting protein, L23, is more negatively charged than L29. Yet, it bears a solvent-exposed nonpolar region that is more evident in the $±10$ $k_BT/e$ image of Fig. 6e. Moreover, Fig. S10 also shows that much of the negative charge of the solvent-exposed region of L23 is contributed by this protein's C terminus, which is expected to be deprotonated near neutral pH ($pKa ≤ 4.03$)[65], but may be highly dynamic[66]. Hence, a portion of the RNC-ribosomal-protein interactions may be of nonpolar nature.

A ca. 30% decrease in crosslinking fraction was found upon switching from PIR 1.0.3 to PIR 1.2.3 (Fig. 6c). The K23E and R25E mutations responsible for the generation of PIR 1.2.3 may lead to the disruption of specific Coulombic interactions involving the K23 or K23/R25 residues. Alternatively, these mutations may cause disruption of interactions via indirect effects. For instance, an increased PIR-1.2.3 chain dynamics induced by a high level of intra-chain repulsion in this mutant (bearing 12 negatively charged residues) may lead to an increased chain entropy and result into the disruption of interactions of any kind, anywhere within the PIR-1.2.3 chain. Figure 5f shows the mean fractional buried area[67] of N-terminal regions of PIR RNCs. The plot regions joining the K23 to the R25 residues correspond to highly polar portions of both the PIR 0.0.0 and PIR 1.2.3

chains. Hence, varying the PIR amino-acid sequence from the 0.0.0 to the 1.2.3 variant did not cause any changes in chain polarity, suggesting that direct changes in hydrophobicity do not account for the differences in the extent of crosslinking experienced by these two RNCs. As discussed above, indirect effects cannot be ruled out.

In summary, our data show that PIR RNCs are overall insensitive to large variations in net negative charge, suggesting that conventional Coulombic effects are unlikely to play a dominant role in RNC-ribosomal-protein interactions. Other types of interactions must also be present. These may include metal-cation- or water-mediated negative-charge-to-negative-charge electrostatic interactions[68] or interactions of non-electrostatic nature (e.g., hydrophobic-type effects). The effect of $Mg^{+2}$ ions is explicitly explored two sections below.

**Interaction patterns depend on RNC chain length and amino-acid distribution.** To evaluate the contribution of amino-acid position along the nascent chain, we explored the interactions experienced by shorter (70 residues) incomplete-length PIR RNCs. The polar/nonpolar content profile of nascent $PIR_{70}$, highlighting the position of all charged residues (including the EDC-reactive Ds, Es, and Ks), is plotted in Fig. S11a. Figure S11b shows the amino acid sequence of $PIR_{70}$ 0.0.0 (WT), 1.0.3, and 1.2.3 and the outcome of crosslinking experiments involving all three versions of PIR RNCs. Interestingly, all these RNCs exhibit identical crosslinking behavior, within error, and an overall decreased fraction of crosslinking relative to the full-length RNCs of Fig. 6c. Note that, for $PIR_{70}$, the difference in the interactions upon switching from the 0.0.0 to the 1.2.3 variant, which was detected for $PIR_{91}$ (compare PIR 1.0.3. and 1.2.3 RNCs in Figs. 6c and S11), is no longer present. We conclude that RNC-ribosomal-protein interactions must depend on both RNC chain length and amino-acid distribution.

**Ribosome-nascent chain interactions are mostly mediated by $Mg^{+2}$ ions.** In order to assess whether divalent cations play a role in ribosome-nascent-chain interactions, we examined the PIR RNC solutions at variable concentrations of $Mg^{+2}$ (added to the resuspension buffer). As shown in Fig. 7b, a very significant decrease in fractional crosslinking was observed for all RNCs except for PIR 1.2.3, as the $Mg^{+2}$ concentration of the resuspension buffer was lowered from 10 to 1 mM ($P < 0.05$, two-tailed $t$-test). The activity of the 50S ribosomal subunit at 1 and 10 mM added-$Mg^{+2}$ concentration was confirmed via a puromycin assay on WT $PIR_{91}$ RNCs (Fig. 7b)[38]. Note that 1 mM $Mg^{+2}$ lies within the well-established range of 0.8–2.2 mM range for intracellular free-$Mg^{+2}$ concentration in the $E.$ $coli$ cytoplasm[69,70]. Therefore, our 1 mM added-$Mg^{+2}$ concentration (see Fig. 7b, lane 5 of Gel 3) can be regarded as falling within the physiologically relevant regime, where the ribosome is active and most likely also structurally intact. Interestingly, Fig. 7c shows that the crosslinking fraction decreases sharply between 1.5 and 1 mM added-$Mg^{+2}$, for WT $PIR_{91}$ ($P < 0.05$, two-tailed $t$-test). Therefore, we conclude that $Mg^{+2}$ ions mediate a large fraction of the PIR RNC interactions with ribosomal proteins under physiologically relevant conditions. Note that L23, the major interacting component identified in this work, is an essential protein for $E.$ $coli$ cell viability[71]. Therefore L23 is most likely ribosome-associated within the entire physiologically relevant $Mg^{+2}$ concentration of 0.8–2.2 mM.

Interestingly, Fig. 7b-c shows that PIR 1.2.3 RNCs (net charge = −10.9) displays a lack of dependence on $Mg^{+2}$ concentration ($P > 0.05$, two-tailed $t$-test), unlike all other tested nascent chains. This result suggests that nascent proteins bearing

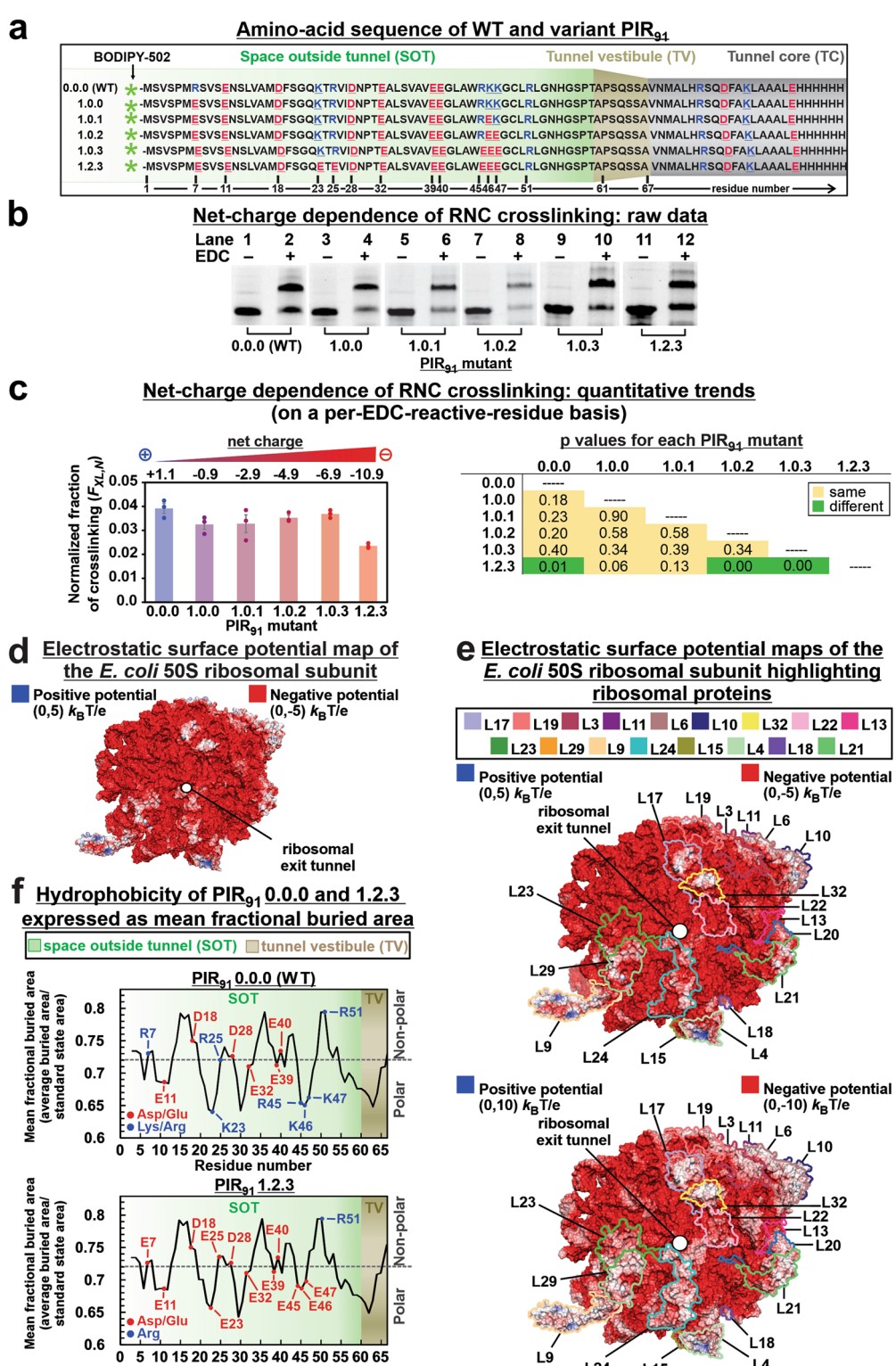

**Fig. 6 Net-charge dependence of fraction of crosslinked PIR RNCs. a** Schematic representation of amino acid sequence of PIR$_{91}$ RNCs WT and variants with acidic and basic residues, shown in red and blue, respectively. EDC-reactive residues are underlined. The space outside tunnel (SOT), tunnel vestibule (TV), and tunnel core (TC) is color-coded. **b** Representative SDS-PAGE of PIR RNCs variants and EDC-exposed RNCs. **c** Normalized crosslinking fractions determined as in Fig. 5b. Error bars denote ±SE for $n = 3$. The $p$ values from two-tailed $t$-test ($P < 0.05$) are tabulated. **d** Electrostatic surface potential map of the 50S subunit of ribosome from *E. coli* (PDB ID: 4YBB) in proximity to the exit tunnel. See Methods section for details on surface-potential determination. **e** Electrostatic surface potential map of the 50S *E. coli* ribosomal subunit (PDB ID: 4YBB) highlighting the ribosomal proteins, determined as in **d**. **f** Hydrophobicity scores of WT PIR$_{91}$ (0.0.0, upper plot) and PIR$_{91}$ 1.2.3 (lower plot) nascent-chain residues located in the SOT and TV according to Model I of Fig. 5f. Hydrophobicity was expressed as mean fractional buried area according to Rose et al.[67] The scores were obtained via the Expasy server using a residue window size of 5.

a large negative net charge are able to interact with $Mg^{+2}$ even when the bulk concentration of this ion is very low. This phenomenon may be facilitated by the abundance of RNC negative charges, which can favor $Mg^{+2}$ binding by providing extra metal-coordinating capacity.

The block diagram in Fig. 7d highlights the effect of decreasing $Mg^{+2}$ concentration on ribosomal-protein interactions involving PIR nascent chains of different length, composition, net charge, and hydrophobicity. The dependence on $Mg^{+2}$ concentration is proportional to the $\Delta F_{X,N,\ Mg+2}$ parameter. The role of different nascent-chain characteristics is clearly complex and hard to pinpoint with certainty without any additional detailed studies. Nonetheless, the available trends suggest that longer and more negatively charged RNCs may generally be less sensitive to variations in $Mg^{+2}$ concentration, potentially due to a larger number of $Mg^{+2}$-binding-competent functional groups (e.g., carboxylates). Future investigations will target a systematic verification of this hypothesis. In general, it is clear that $Mg^{+2}$-mediated interactions between RNCs and ribosomal proteins are RNC-sequence-dependent.

## Discussion

This study provides the first direct evidence for interactions between RNCs lacking signal or translational-stalling sequences and a well-defined region of the ribosomal surface in the vicinity of the ribosomal exit tunnel. A significant fraction of full-length RNCs of the intrinsically disordered protein PIR crosslinks with ribosomal protein L23 and, more weakly, with the L29 protein, in a chaperone-independent fashion (Figs. 1d, 3, 4, and S1). Remarkably, the interacting proteins are clustered on one specific side of the ribosomal tunnel vestibule and ribosomal outer surface in the vicinity of the nascent-chain tunnel exit site (Fig. 4c–e).

The crosslinking strategy employed in this work is unable to quantitatively assess the populations of interacting and non-interacting RNCs. On the other hand, comparisons with the independently assessed fractions of conformationally biased nascent PIR via fluorescence anisotropy decays[21] was exploited to validate our findings and, in addition, to provide reliable population assessments. Hence we propose the merging of chemical crosslinking and fluorescence anisotropy decay as a powerful synergistic approach to qualitatively and quantitatively assess interactions of nascent chains with the ribosomal surface.

Surprisingly, most of the interacting proteins associate with the nascent chains even at shorter lengths, between 50 and 91 residues (Fig. 5a, b), suggesting a pervasive role for these interactions. The extent of interactions of PIR RNCs on a per-residues basis increases with RNC chain length (Fig. 5a, b). This result suggests that longer chains sample a wider ribosomal-protein surface and therefore experience an overall stronger affinity for the ribosome. It is worth stressing that the scarce extent of detected crosslinking experienced by very short RNCs (30 and 40 residues) may either be due to reduced interactions with ribosomal proteins or to low EDC accessibility to the exit tunnel core.

Based on the observations summarized below, we propose two structural models (Models I and II), illustrated in Fig. 5f, for the ribosomal-surface sampling of the intrinsically disordered PIR nascent chain. First, we detected a change in slope in cone semi-angle of the bound N-terminal residues after about 29 residues have traversed the tunnel (observation 1, Fig. 5d). Second, we observed a clear correlation between the fraction of crosslinked RNCs and the number of EDC-reactive residues on either the outer ribosomal surface or the ribosomal vestibule (observation 2, Fig. 5e).

Interestingly, the interactions of nascent PIR with the ribosomal surface are not dominated by classical electrostatic forces

involving direct contacts between negatively charged ribosomal surface and positively charged nascent-PIR side chains (Fig. 6). On the other hand, some of these electrostatic interactions, perhaps involving K23 and R25, may be present.

The particularly large nonpolar solvent-exposed surface of L23 (Fig. 4f), the known unfolded or partially folded nature of native chains[21,22,36,72], and the data of Fig. 7b suggest that both hydrophobic effects and $Mg^{+2}$ play a role in the detected interactions.

Notably, we found that $Mg^{+2}$ mediates a significant fraction of the interactions between PIR RNCs and ribosomal proteins (Fig. 7b–d). The $Mg^{+2}$ ions known to be associated with the intact empty ribosome (i.e., lacking nascent chains) can be either site-bound or diffuse[73,74]. The site-bound $Mg^{+2}$ ions[75] exhibit low mobility, reside close to the PTC[73,76] and stabilize the ribosomal structure via charge transfer, polarization, and exchange correlation[73]. Diffuse ions are more abundant and comprise hydrated $Mg^{+2}$ which retains fairly-high mobility. Diffuse ions are also known to stabilize RNA and RNA-protein complexes[73,74]. Fig. 8a shows two qualitative proposed limiting models for the potential $Mg^{+2}$-mode of binding in the context of the $Mg^{+2}$-mediated ribosome-nascent-chain interactions. Due to the low-density of site-bound $Mg^{+2}$ ion away from the PTC[73,77], and given the high energetic cost of atom displacement from the first-shell of $Mg^{+2}$ ions[74], site-bound (chelated) $Mg^{+2}$ may not be the favorite binding mode. We note that diffuse $Mg^{+2}$ ions are also conceptually more viable due to their ability to support a fluid interaction network and RNC conformational reorganization during translation elongation.

In summary, this work unravels a new role of the ribosome by showing that specific regions of the 50S subunit actively engage in interactions with the nascent protein chain even in the absence of signal or arrest sequences. The key features of our findings are schematically illustrated in Fig. 8b. Based on our data, we propose that the detected ribosomal-protein-RNC interactions may play a role in RNC conformational channeling and in maintaining high nascent-chain solubility during translation. Future studies exploring other nascent protein sequences and RNA-nascent-protein interactions will broaden the scope of our findings and unveil their impact on protein biosynthesis and folding.

## Methods

**Gene sources**. Wild-type GRB14 PIR gene with C-terminal His6 tag and stop codon mutated to GAT in pET-28b vector was obtained from Isabelle Broutin (Laboratorie de Cristallographie et RMN Biologiques, Paris, France). Mutations to produce PIR variants were introduced via the Quikchange™ mutagenesis kit (Stratagene) either in house or by Genscript. Codons of relevant positively charged residues were replaced with glutamic acid codons of high E. coli usage frequency.

**Generation of BODIPY-502-labeled RNCs**. Ribosome-nascent chain complexes (RNCs) were produced using an in-house cell-free coupled transcription-translation system prepared from A19 E. coli and purified as described[22]. In brief, cell-free reactions were incubated at 37 °C in the presence of BODIPY-502-Met-tRNA$_{fMet}$ and allowed to react for 30 min. Unnatural BODIPY-502-Met-tRNA$_{fMet}$ was prepared as described[22]. Stalling of ribosomes to create PIR RNCs of 30, 40, 50, 50, 70, 80, and 91 amino acids was achieved through the use of the appropriate DNA oligonucleotides at 0.15 μg/μL concentration to promote the deoxyribonucleotide-directed mRNA cleavage of C-terminal stop codon via RNase H[30]. To prevent ribosomal rescue of stalled RNCs, 0.15 μg/μL anti-SsrA oligonucleotide was added to the cell-free mixture prior to translation initiation[78]. RNCs were isolated via ultracentrifugation at 68,000 RPM through a solution containing 1.1 M sucrose, 20 mM tris base, 10 mM Mg(OAc)$_2$, 500 mM NH$_4$Cl, and 0.5 mM EDTA, pH 7.0 at 4 °C for 1 h. The ribosomal pellet was dissolved in resuspension buffer (10 mM tris-HCl, 10 mM Mg(OAc)$_2$, 60 mM NH$_4$Cl, 0.5 mM EDTA, and 1.0 mM DTT, pH 7.0) by shaking at 250 RPM for 1 h on ice. The crosslinking reaction dependence on KCl or $Mg^{+2}$ was assessed by dissolving the ribosomal pellet in resuspension buffer at the indicated concentrations of either KCl or Mg(OAc)$_2$ and samples were incubated for 1 h prior EDC-exposure. The ribosome-bound nature of the peptidyl-tRNA specie was assessed by exposure to 1 mM puromycin at 37 °C for 30 min. All RNCs were analyzed in low-pH SDS-PAGE[35]. Gels were imaged with an FLA 9500 Typhoon Gel Imager with a 473 nm excitation laser and a BPB1 emission filter to detect the fluorescent signal of BODIPY-502,

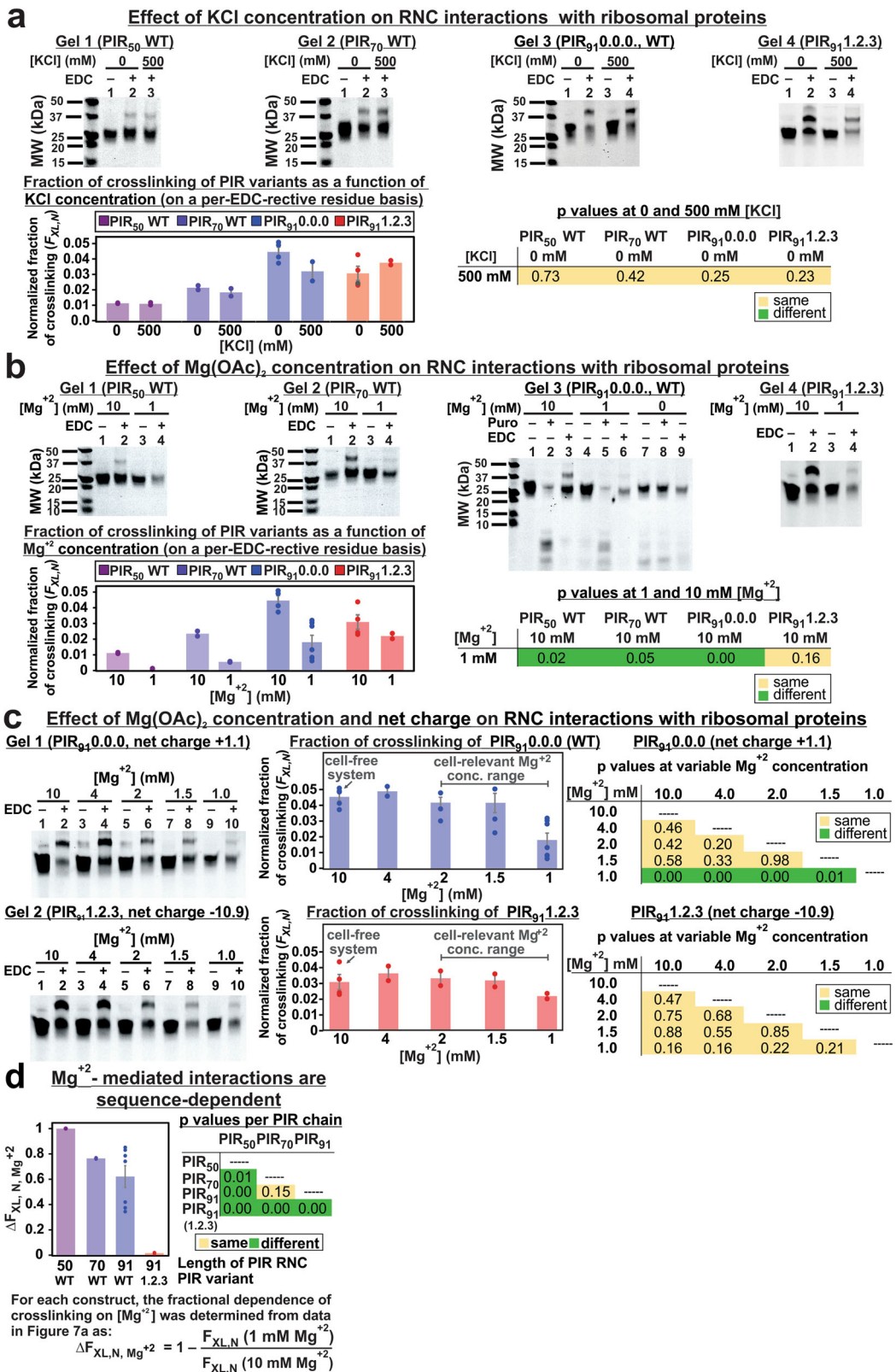

**Fig. 7 Effect of ionic strength and Mg$^{+2}$ concentration on PIR nascent-chain crosslinking. a** Representative gels and normalized crosslinking fractions of nascent PIR chains of increasing length (PIR$_{50}$, PIR$_{70}$, and PIR$_{91}$) and variable net charge (PIR$_{91}$ 1.2.3). RNCs were resuspended in buffer at 0 or 500 mM KCl. Error Bars denote ±SE for $n = 2$–4. **b** Effect of Mg$^{+2}$ concentration on the fraction of crosslinked PIR variants. WT PIR nascent chains of increasing length (PIR$_{50}$, PIR$_{70}$, and PIR$_{91}$) and PIR$_{91}$ 1.2.3 RNCs were resuspended in buffer containing 1 or 10 mM Mg$^{+2}$. Error bars denote ±SE for $n = 2$–7. **c** Effect of gradual variations in Mg$^{+2}$ concentration on the nascent-chain crosslinking of PIR$_{91}$ 0.0.0 and PIR$_{91}$ 1.2.3. Error bars denote ±SE for $n = 2$–7. **d** Normalized fractional dependence of crosslinking of PIR$_{50}$, PIR$_{70}$, and PIR$_{91}$ 0.0.0. on Mg$^{+2}$ concentration ($\Delta F_{XL,N,Mg^{2+}}$). All $\Delta F_{XL,N,Mg^{2+}}$ values were determined from data in Fig. 7b via the equation shown in the panel. Likewise, normalized crosslinking fractions in panels **a** and **c** were obtained as in Fig. 5b. All $p$ values from two-tailed $t$-tests ($P >$ or $< 0.05$) are shown. Samples were incubated at the listed salt concentrations for 1 h prior to EDC crosslinking.

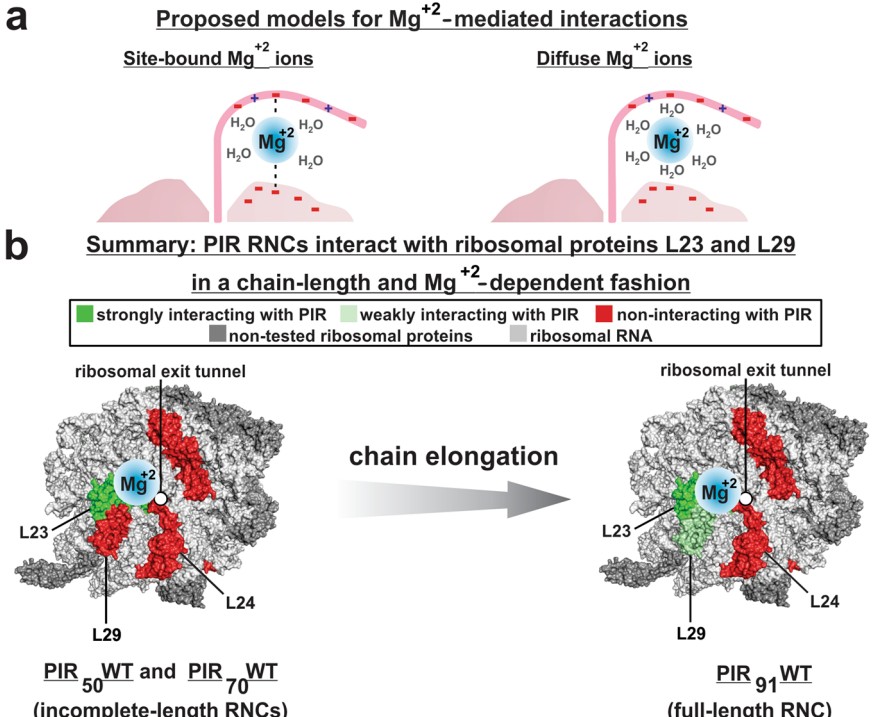

**Fig. 8 Overview of interactions between nascent chains (RNCs) and ribosomal proteins identified in this study. a** Proposed limiting models for $Mg^{+2}$-mediated nascent-chain-ribosomal-surface interactions. **b** Schematic representation of the major features of nascent-chain-ribosomal-protein interactions detected in this work. Briefly, L23 is the major interacting protein though L29 also experiences weak interactions with longer PIR nascent chains. A considerable fraction of the interactions is mediated by $Mg^{+2}$ ions.

while a 635 nm excitation laser was used to image the molecular-weight markers. The combined molecular weight of PIR nascent chains (91 residues, ~9 kDa) and their covalently linked tRNA (~28 kDa) is ca. 37 kDa. However, PIR RNCs migrate slightly above the apparent 25 kDa line (Fig. 1d, lane 1) due to the presence of residual tRNA tertiary structure, under the running conditions of the gel.

**Generation of crosslinked RNCs**. Resuspended RNCs were aliquoted and kept on ice prior to the crosslinking reaction. The EDC crosslinker ((1-ethyl-3-(3-dime-thylaminopropyl)carbodiimide hydrochloride), Thermo Fisher Scientific) stock was prepared at 800 mM using DNase/RNase free water (Corning) and the pH was adjusted to 6.8–7.0 with 1 M KOH. In all, 0.66 μL of the EDC stock was then added to 6 μL of the re-suspended RNCs to give a final concentration of 80 mM and the reactions were incubated at 30 °C for 30 min. Crosslinking reactions were promptly quenched upon quick addition of 0.74 μL of quenching buffer (QB: 1 M Tris-HCl pH 7.0, 1 M glycine, and 1 M KOAc) and placed on ice. After quenching, the reactions were stored at 4 °C for later SDS-PAGE analysis.

**Assessment of crosslinking fractions**. The fraction of crosslinking for PIR$_{91}$, PIR incomplete chains and PIR variants was determined upon evaluating the intensity of the crosslinked and non-crosslinked RNC bands with the ImageJ[79,80] software. Examples of these quantifications are provided in Fig. 2a. In short, the entire lane of the gel was selected, plotted, and the areas of the crosslinked and non-crosslinked peaks were quantified. After plotting the profile of a particular lane, the background was manually subtracted via the rolling ball method (see ImageJ manual). Individual peaks pertaining to the crosslinked and non-crosslinked band were defined using the line tool. Intensities were then quantified. The fraction of crosslinking ($F_{XL}$) was determined according to the expression

$$F_{XL} = \frac{XL_{RNCs}}{XL_{RNCs} + non - XL_{RNCs}}, \qquad (2)$$

where $XL_{RNCs}$ and $non-XL_{RNCs}$ denote fluorescence-detected gel band intensities of the crosslinked and non-crosslinked RNCs, respectively. The normalized cross-linking fraction ($F_{XL,N}$) was calculated by dividing the fraction of crosslinking ($F_{XL}$) by the total number of EDC-reactive residues per RNC.

**Overexpression and purification of TF and Hsp70 chaperones**. The *E. coli* Hsp70 (i.e., DnaK) was overexpressed and purified as described[81,82].

Trigger factor (TF) was prepared as follows. The wild-type *tig* gene (encoding *E. coli* TF) was subject to PCR amplification and subcloned into the linearized Champion™ pET SUMO vector (Invitrogen, Carlsbad, CA) to produce the His-TF-pET-SUMO plasmid. Note that the Champion™ pET SUMO vector encodes an N-terminal His tag preceding the SUMO-encoding nucleotides. BL21CodonPlus(DE3)-RIPL (Agilent Technologies, Palo Alto, CA) competent cells were then transformed with His-TF-pET-SUMO and grown in Luria broth (LB) in the presence of 15 μg/mL kanamycin and 34 μg/mL chloramphenicol at 37 °C. His-SUMO-TF overexpression was induced by addition of 1 mM IPTG at 0.7–0.8 $OD_{600nm}$. Cells were incubated overnight under shaking (250 r.p.m.) at 25 °C. After induction, cells were harvested, resuspended, and sonicated in a buffer containing 30 mM potassium phosphate (pH 7.4), 500 mM KCl, 20 mM imidazole, and 2 mM phenylmethanesulphonyl fluoride (PMSF). The cell lysate was centrifuged for 20 min at 10,000 rpm [30,260 RCF] (JA-25.50, Avanti JXN-26, Waltham, MA). The supernatant was applied to a HisTrap HP column (GE Healthcare, Piscataway, NJ) and eluted with a buffer composed of 30 mM potassium phosphate (pH 7.4), 500 mM KCl, and 500 mM imidazole. The fractions containing His-SUMO-TF were dialyzed against 30 mM potassium phosphate (pH 7.4), 500 mM KCl, and 20 mM imidazole. Then, 5000 units of His-tagged SUMO protease (SP-400, MCLAB, South San Francisco, CA) per 1 μmol of purified His-SUMO-TF were added, followed by overnight incubation at 30 °C. Samples were loaded on a HisTrap HP column (GE Healthcare, Piscataway, NJ) and the flow-through was collected and verified (by SDS-PAGE) to include exclusively wild-type TF and no His-SUMO-TF, His-SUMO tag, or His-tagged SUMO protease. Finally, size exclusion chromatography was performed on a Superdex 200 column (GE Healthcare, Piscataway, NJ) with an elution buffer composed of 50 mM Tris (pH 7.2), 5 mM MgCl2, 50 mM KCl. Wild-type TF-containing fractions were collected, concentrated, flash-frozen, and stored in 50 mM Tris (pH 7.2), 5 mM MgCl$_2$, 50 mM KCl at −80 °C.

**Ligand-binding titrations to assess the affinity of PIR RNCs for TF and K/J/E chaperones**. Individual samples of resuspended PIR$_{91}$ RNCs were incubated on ice for 10 min in the presence of increasing concentration of TF or mixtures of DnaK (K), DnaJ (J), and GrpE (E) (K/J/E) chaperones. Solutions were then treated with the crosslinking agent EDC (80 mM total concentration) and incubated for 30 min at 30 °C. TF ligand-binding titrations were carried out in the standard resuspension buffer (10 mM tris-HCl, 10 mM Mg(OAc)$_2$, 60 mM $NH_4Cl$, 0.5 mM EDTA, and 1.0 mM DTT, pH 7.0), while K/J/E titrations were performed at K/J/E 5:1:2 molar ratios in the presence of 5 mM ADP, 50 mM KCl, and a modified version of the resuspension buffer containing 5 mM $Mg^{+2}$. The presence of 50 mM ADP ensured generation of the DnaK ADP-bound state (ADP-DnaK).

**Analysis of crosslinking titration data**. The interaction of $PIR_{91}$ RNCs with either TF or ADP-DnaK was modeled as described[83], as a simple two-state process

$$P + C \rightleftharpoons PC, \qquad (3)$$

where P, C, and PC denote $PIR_{91}$ RNCs, TF, or DnaK-ADP chaperones, and the $PIR_{91}$-RNC chaperone complex, respectively. The apparent dissociation constant ($K_{d,app}$) corresponding to the above process can be expressed as

$$K_{d,app} = \frac{P_{eq}C_{eq}}{PC_{eq}}, \qquad (4)$$

where $P_{eq}$, $C_{eq}$, and $PC_{eq}$ denote the equilibrium concentrations of $PIR_{91}$ RNCs, TF or ADP-DnaK chaperones, and the $PIR_{91}$ RNC−chaperone complex, respectively. $K_{d,app}$ can also be expressed as

$$K_{d,app} = \frac{(P_0 - fbP_0)(C_0 - fbP_0)}{fbP_0}, \qquad (5)$$

where $P_0$ and $C_0$ denote the total concentrations of $PIR_{91}$ RNCs and TF or DnaK-ADP chaperones, respectively, and $f_b$ is the fraction of $PIR_{91}$ RNCs bound to TF or ADP-DnaK. Equation 5 can be solved for $f_b$. The only physically meaningful solution is

$$f_b = \frac{K_{d,app} + P_0 + C_0 - \sqrt{(K_{d,app} + P_0 + C_0)^2 - 4P_0C_0}}{2P_0}, \qquad (6)$$

Finally, relation 6 was used to fit the binding-isotherm data (see Fig. S1b), where $f_b$ was regarded as equivalent to either $F_{XL-TF}$ or $F_{XL-DnaK}$. Nonlinear least-squares data fitting was carried out with the Origin software package (version 2019b).

**Kinetic studies to assess the timecourse of crosslinking**. Purified RNCs were generated and kept on ice prior to the crosslinking reaction. Ten identical aliquots (6 μL each) were incubated at 80 mM EDC final concentration at 30 °C. After each of the time intervals of interest (0.1, 1, 5, 7.5, 10, 15, 20, 25, 30, and 40 min), each one of the aliquots was quenched upon rapid addition of 0.74 μL of quenching buffer (QB) at 4 °C and was immediately placed on ice. RNCs were analyzed by low-pH SDS-PAGE and quantification of gel-bands intensities was carried out with the Image J software package. The crosslinking kinetic experiments were carried out in duplicate (Fig. S2).

**Analysis of crosslinking kinetic data**. The experimental kinetic data were fit to the kinetic model of Fig. 2b, disregarding the proposed burst-phase steps, via the Copasi kinetic-simulation software (version 4.22)[84,85]. The model involves a slow conformational step, described by the first-order rate constant $k_1$, followed by a fast step accounting for the second stage of crosslinking and characterized by $k_2$. The $k_2$ value was fixed to $1 \times 10^9 \, s^{-1}$ and regarded as a unimolecular step (within the ribosome-nascent-chain complex) characterized by a 1st-order rate constant. In order to maximize accuracy, 800,000 iterations were carried out for the fitting of each experimental data set. The parameters $k_1$ and initial protein concentration were allowed to float, and the fraction of crosslinking was converted to molar concentration of carboxyl groups in PIR RNCs and ribosomal proteins L23 and L29 (total number of carboxyl groups = 29) multiplied by the ribosome concentration of resuspended RNCs (40 nM). Reduced chi-squares ($\chi_R^2$) were computed accordingly to the relation[86]

$$\chi R^2 = \frac{1}{2nf - 1} \sum \frac{Y_e - Y_c}{\partial Y}, \qquad (7)$$

where n is the number of experimental points, f is the number of free-floating parameters in the curve fit (f = 2 in this study), $Y_e - Y_c$ are the experimental and calculated data, respectively, and $\delta Y$ is the uncertainty associated with the experimental measurement. The $\delta Y$ parameter was estimated to be 0.083. This value was derived from the standard error for 20 independent crosslinking reactions performed under the same experimental conditions.

**Origin and validation of antibodies**. Aliquots of the rabbit anti-uL23 antibody were kindly donated by Shu-ou Shan (California Institute of Technology). The anti-uL23 antibody was generated by GenScript, using the CGKVKRHGQRIGRRS peptide as epitope and has been previously validated[87]. Rabbit anti-uL17, -uL18/L22, -uL24, -uL29, and -uL32 antibodies were kindly facilitated by Bryan W. Davies (University of Texas-Austin) and Melanie Oakes (University of California, Irvine) who obtained them from Masayasu Nomura (University of Wisconsin-Madison). These antibodies were generated using the purified *E. coli* ribosomal proteins L17, L23, L18+L22, L24, L29, and L32[88]. The specificity of each antiserum was assessed via the Ouchterlony immunodiffusion assay[89] against all ribosomal proteins[88]. These antibodies were successfully used in previous published studies[88,90].

**Identification of ribosomal proteins crosslinked to nascent PIR via western blotting**. Resuspended RNCs, puromycin released RNCs, crosslinked RNCs, puromycin released crosslinked RNCs, and puromycin released RNCs subsequently crosslinked were separated via low pH SDS-PAGE[35], then transferred to a PVDF membrane. Membranes were probed with rabbit anti- *E. coli* L23, uL24, uL29,

uL18, uL32, uL17, and uL22 polyclonal antibodies at a dilution of 1:2000 overnight at RT on a rocking platform (incubation of 1 h gave identical results), washed with TBST and further exposed to goat anti-rabbit secondary antibodies. In the experiments of Figs. 3a–c, f and S3a–b, detection was carried out via chemiluminescence (CDP-STAR substrate, Millipore Sigma) after sample incubation for 1 h at RT on a rocking platform using a 1:10000 secondary antibody dilution (Novagen). In the experiments of Figs. 3d, e, S3c, S3d, S3e, and S3f, a 1:20,000 secondary antibody dilution was used (Alexa Fluor Plus 647 Thermo Fisher), and samples were incubated for 45 min at RT on a rocking platform followed by fluorescence detection using a 635 nm excitation laser. Note that anti-uL24 antibodies (Fig. 3d) displayed minor bands on western blots due to dimer formation (L24 monomer MW = 11.2 kDa), anti-uL29 antibodies (Fig. 3e) displayed bands due to dimers, tetramers and larger oligomers (L29 monomer MW = 7.3 kDa), and anti-L32 antibodies (L32 monomer MW = 6.3 kDa) displayed mostly a band due to a tetramer (Fig. 3f). The SDS-PAGE behavior of some *E. coli* ribosomal proteins as apparent multimers had been previously observed[91] and is not entirely surprising, given the charge segregation of ribosomal proteins from different organisms including *E. coli* which promotes strong association between them[59].

**Computation of electrostatic surface potential**. Protein Data Bank files from the 4YBB structure were converted to PDB2PQR files and used as input for electrostatic potential calculations using the Adaptive Poisson-Boltzmann Solver (APBS) software package for PyMOL, developed by Baker et al.[92]. The linearized Poisson-Boltzmann equation was used to perform electrostatic surface calculations[93]. Parameters for these calculations were as described by Fedyukina et al.[59] (150 mM ionic strength, solute dielectric of 2.0, and solvent dielectric of 78.0) The ribosome with Protein Data Bank ID 4YBB was employed. Regions with positive and negative surface potential are displayed in blue and red, respectively. $k_BT$ denotes an energy unit of $4.11 \times 10^{-21} J$ at room temperature, where $k_B$ is the Boltzmann constant (in J/K), T is the temperature (in K), and e denotes the electric charge (in Coulomb).

**Statistics and reproducibility**. Statistical data analysis was performed with the Excel software. Data are displayed as mean ± standard error (SE), with the number of independent experiments listed in parenthesis, (e.g., n = 3). Statistically meaningful differences between pairs of data were assessed via the two-tailed Student t test. Pairs of results were regarded as statistically different if bearing P values <0.05.

**Reporting summary**. Further information on research design is available in the Nature Research Reporting Summary linked to this article.

## Data availability
The source data for the graphs and charts are available as Supplementary Data 1 and any remaining information can be obtained from the corresponding author upon reasonable request.

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

## Acknowledgements

We are grateful to Rayna Addabbo and Matt Dalphin for helpful discussions and for gifts of small amounts of *Δtig E. coli* cell strain and Hsp70 chaperone. In addition, we thank Shu-ou Shan for donating the antibodies against the *E. coli* ribosomal protein L23, Bryan W. Davies and Melanie Oakes for donating antibodies against the *E. coli* ribosomal proteins L17, L18/L22, L24, and L29. V.G.-L. and A.M.F. were grateful recipients of a postdoctoral scholarship by CONACyT and a Molecular Biophysics Fellowship by NIH, respectively. This work was funded by the National Science Foundation (NSF) grants MCB-1616459, MCB 2124672, and CBET-1912259 to S.C.

## Author contributions

V.G.L. and A.M.F. designed and performed the experiments, analyzed the data, including statistical analysis, and wrote the manuscript. In addition, V.G.L. performed the kinetic and binding-isotherm data fitting/analysis, and carried out the kinetic simulations. A.J.A. performed the electrostatic surface potential calculations and generated all the related figures and videos. S.C. designed the project, participated in data analysis, wrote the manuscript, and contributed to manuscript editing. A.S. expressed and purified the TF chaperone and reviewed manuscript drafts.

## Competing interests

The authors declare no competing interests.
