## [Peer Review File · Communications Biology]

Reviewers' comments:

Reviewer #1 (Remarks to the Author):

The manuscript by Guzman-Luna presents a careful crosslinking analysis of the interaction of unstructured nascent chains with ribosomal proteins near the nascent polypeptide exit tunnel of the ribosome. The study is timely, as there is emerging data that the ribosome can exert a profound influence on the folding of emerging nascent proteins during translation; however, the ribosomal proteins that provide potential interaction surfaces for nascent proteins have not been systematically identified/characterized. In this work, the authors used crosslinking and western blot analyses to show that an intrinsically disordered nascent protein can contact multiple ribosomal proteins, such as L23, L29 and potentially L24, which cluster on one side of the ribosome surface at the exit tunnel, and that the probability of the interactions increase as the nascent chain elongates. While the functional implications of these interactions still need to be determined, this is an important and interesting piece of information for the field and could facilitate the understanding of cotranslational protein folding and maturation in the long-term.

While these observations are clear and convincing, the manuscript tries to extract too much information from limited data, resulting in over-interpretation of data in multiple instances that distract from the main message. Some of these additional major conclusions, such as the electrostatic nature of nascent chain-RNC interactions, should be supported by additional data. Specifics of these comments are detailed below.

1. The kinetic analysis of the EDC crosslinking data: While I agree with the authors that the time course of the crosslinking reaction is biphasic, the data do not warrant the kinetic fitting described in Figures 1 and 2. For example, only two data points (at 0.1 min and 1 min) are on the same timescale as one of the fitted kinetic parameters ($\tau_{\text{for}} = 70\text{s}$), and both crosslinking rate constants (τ_1 and τ_2) are too fast for any of the experimental time points. Moreover, the fitted rate constants did not reproduce the experimental data when simulated. This is a typical case of overfitting. An empirical fitting of the data with two net forward rate constants would be more appropriate, and the authors should stay with the simplest interpretation (a fraction of nascent chains crosslink quickly with the ribosome while the rest do so slowly). The nature of the slowly crosslinking NC species was never clarified in the manuscript, and fraction and reactivity of this population seems to vary substantially from experiment to experiment; as such, the authors should refrain from interpreting this population in the Discussion.
2. While the crosslinks of NC to L23 and L29 is convincing (Figure 3), the crosslinks to L24 is much less clear due to high background in the lane. This data needs to be repeated.
3. The authors constructed a series of charge-reversed mutations on the PIR nascent chain and, in most cases, did not observe major changes in crosslinking efficiency. It wasn't until residues 23-25 were mutated that the authors observed a reduction in crosslinking efficiency. The authors concluded from these data that Coulombic interactions are involved in NC interactions. However, alternative interpretations of the data are not considered, and the correction of crosslinking efficiency by the number of E/D residues is not justified by data, as the author later showed that the position of residues in the nascent chain is also important for crosslinking efficiency. To establish the importance of electrostatic interactions, dependence of the crosslink on ionic strength will be helpful. In addition, mutations that reduce charge, rather than increase/reverse charge as well as changing the number of crosslink-reactive residues, need to be tested. Finally, to establish that residues 23-25 make specific interactions with a basic site on the ribosome surface, these residues need to be mutated in the context of WT PIR instead of the hyper-mutated variant.
4. Presentation of the electrostatic surface potential of the ribosome in Figure 6d is confusing. Is the negatively charged surface dominated by ribosomal RNA, or is it contributed by proteins? Further, the color scaling of APBS in this figure seems narrow ($\pm 3 \text{ kT/e}$) and may artificially accentuate negative charge. The figure gives the impression that the ribosome surface surrounding the exit site is highly acidic and drove the model in Discussion that NC interactions with the ribosome are electrostatic; however, this conclusion runs counter to the observation that extensively changing the charge density and nature of PIR nascent chain has no effect on crosslinking efficiency.
5. Based on the SDS-PAGE running pattern of NC crosslinks, the authors concluded that the bands represent simultaneous crosslinks of the PIR NC to multiple ribosomal proteins. Additional evidence

is needed for this conclusion. The migration of crosslinked protein complexes in SDS-PAGE can vary substantially depending on the site of crosslink and conformation of the crosslinked complex, without invoking the presence of additional crosslink partners.

Reviewer #2 (Remarks to the Author):

The manuscript "Interaction of an intrinsically disordered nascent protein with specific regions of the ribosomal surface" presents an interesting analysis on the interactions between ribosome-bound nascent chains of an intrinsically disordered protein and the proteins around the ribosomal exit tunnel region. By a cross-linking based approach, the authors detail a relevant pattern of interactions within selected ribosomal proteins, but not others, and the nascent peptide chain. Furthermore, they analyze the dependence of these interactions on the peptide length and its surface charge, using an elegant mutagenesis approach.

I think the points and the conclusions raised in the manuscript are original and sound, and I am convinced it presents an important first step in shedding light in the still relatively unknown universe of ribosomal interactions with peptide nascent chains. I also think the authors do a great job in keeping the text effective and pleasant to read.

As a main suggestion for the authors I would encourage them to add more "specificity" to the title of their manuscript, for instance mentioning that the "specific regions of the ribosomal surfaces" are indeed concentrated around the exit tunnel. Also I would find a way to mention that since the study focuses only on protein-protein interactions, keeping protein-rRNA interactions for later.

Additional remarks on two figure legends:

- In the Fig.1 the legend of (a) and (b) panels are swapped.
- In the Fig.4 legend the (e) label is repeated two times and (f) is missing.

Reviewer #3 (Remarks to the Author):

Most of the data are first rate, and except for the suggested mechanism, their interpretation is sound.

I agree with their statement, "The discovery of interactions between nascent proteins lacking signal or arrest sequences and the ribosome is a key step in advancing our knowledge on the influence of the ribosome in protein folding.", but the present work is a modest advance.

The contribution sheds light on what can be done, but, because the protein is intrinsically disordered, the data only dimly illuminate the process. The observations are what one would expect for a disordered protein. In terms of identifying where nascent chains interact with the ribosome, as the authors show in their citation of published efforts, the work is more confirmatory than transformative.

Consider the statement 'this work unravels a new role of the ribosome by showing that specific region of the 50S subunit actively engage in interactions with the nascent protein chain, even in the absence of signal or arrest sequences.' It would be big news if the nascent chain did NOT interact this part of the ribosome.

My key concern is the proposed mechanism. The concentration-versus-time plots in Figure 2c are not warranted given the uncertainties in the rate constants listed in Figure 2b. In several instances the uncertainties in the rate constants are larger than the rate constants themselves, and the confidence intervals includes zero, and negative rate constants. The mechanistic discussion should be stricken.

A problem for the general reader is the lack of detail on stalling. The one sentence on page 19 about the method as it is applied here does not even have a citation.

The manuscript ignores important recent work from the Clark and Shakhnovich labs. Walsh IM, Bowman MA, Soto Santarriaga IF, Rodriguez A, Clark PL: Synonymous codon substitutions perturb cotranslational protein folding in vivo and impair cell fitness. Proc Natl Acad Sci USA 2020, 117:3528.

Jacobs WM, Shakhnovich EI: Evidence of evolutionary selection for cotranslational folding. Proc Natl Acad Sci USA 2017, 114:11434-11439.

as well as recent relevant work from the Christodoulou lab. Waudby CA, Wlodarski T, Karyadi M-E, Cassaignau AME, Chan SHS, Wentink AS, Schmidt-Engler JM, Camilloni C, Vendruscolo M, Cabrita LD, Christodoulou J: Systematic mapping of free energy landscapes of a growing filamin domain during biosynthesis. Proc Natl Acad Sci USA 2018, 115:9744.

The manuscript is longer than it need be. Judicious editing could shorten it by more than 25% Here are several examples of statements that both identify weaknesses in the work and can be deleted.

'Future studies in a more closely in vivo context are needed to confirm the above proposal in living cells.'

'Further analysis based on higher-resolution techniques will shed light on these topics in the future.'

'While not all the species shown may actually be present, the scheme provides an overview of what is possible.'

'Future studies exploring other nascent-protein sequences and employing higher-resolution tools will ascertain the generality of the observed interactions, and will further unveil their role in protein biosynthesis and folding.'

'Interactions of PIR RNCs with rRNA will be addressed elsewhere.'

Abbreviations are inconsistent. Sometimes they are used before they are defined. Other abbreviations are defined more than once.

The reference format, including title capitalization and journal abbreviation is inconsistent.

Silvia Cavagnero
Professor of Chemistry
Department of Chemistry, Room 5357
1101 University Avenue
Madison, WI 53706, USA

Phone: (608) 262-5430
E-mail: cavagnero@chem.wisc.edu

RE: Submission of revised manuscript
COMMSBIO-20-1417-T to *Comm. Biology*

Madison, July 20 2021

Dear Manuscript Reviewers,

I am writing to submit our responses to reviewers' comments for the *Communications Biology* manuscript COMMSBIO-20-1417-T, titled "An intrinsically disordered nascent protein interacts with specific regions of the ribosomal surface near the exit tunnel". For convenience, the changes introduced to the manuscript in response to the reviewers' comments are underlined, in this rebuttal letter. In addition, I also uploaded a version of the manuscript (for review only) with the major implemented text changes shown in red.

I am thankful for the several positive comments about this work. I and the coauthors were glad to address the requested revisions. Our specific responses to each of the reviewers' comments are detailed below.

In response to the comments by Reviewer #1:

Comment 1: The manuscript by Guzman-Luna presents a careful crosslinking analysis of the interaction of unstructured nascent chains with ribosomal proteins near the nascent polypeptide exit tunnel of the ribosome. The study is timely, as there is emerging data that the ribosome can exert a profound influence on the folding of emerging nascent proteins during translation; however, the ribosomal proteins that provide potential interaction surfaces for nascent proteins have not been systematically identified/characterized. In this work, the authors used crosslinking and western blot analyses to show that an intrinsically disordered nascent protein can contact multiple ribosomal proteins, such as L23, L29 and potentially L24, which cluster on one side of the ribosome surface at the exit tunnel, and that the probability of the interactions increase as the nascent chain elongates. While the functional implications of these interactions still need to be determined, this is an important and interesting piece of information for the field and could facilitate the understanding of cotranslational protein folding and maturation in the long-term. While these observations are clear and convincing, the manuscript tries to extract too much information from limited data, resulting in over-interpretation of data in multiple instances that distract from the main message. Some of these additional major conclusions, such as the electrostatic nature of nascent chain-RNC interactions, should be supported by additional data. Specifics of these comments are detailed below.

Response: We are grateful for the careful reading of the manuscript and for the thoughtful and very pertinent suggestions made by this reviewer. We have addressed this reviewer's comments as follows.

Comment 1 (second part): The kinetic analysis of the EDC crosslinking data: While I agree with the authors that the time course of the crosslinking reaction is biphasic, the data do not warrant the kinetic fitting described in Figures 1 and 2. For example, only two data points (at 0.1 min and 1 min) are on the same timescale as one of the fitted kinetic parameters ($\tau_{\text{for}} = 70\text{s}$), and both crosslinking rate constants (τ_1 and τ_2) are too fast for any of the experimental time points. Moreover, the fitted rate constants did not reproduce the experimental data when simulated. This is a typical case of overfitting. An empirical fitting of the data with two net forward rate constants would be more appropriate, and the authors should stay with the simplest interpretation (a fraction of nascent chains crosslink quickly with the ribosome while the rest do so slowly). The nature of the slowly crosslinking NC species was never clarified in the manuscript, and fraction and reactivity of this population seems to vary substantially from experiment to experiment; as such, the authors should refrain from interpreting this population in the Discussion.

Response: We thank the reviewer for these insightful comments on the crosslinking kinetics. Upon further thought, indeed it is now clear that the original model was too complex. We also agree on the fact that the fastest kinetic phase was poorly justified, given the scarcity of available data points at early times. The comments on data overfitting and the kinetic simulations were also excellent. Therefore, to take all the constructive feedback into account, we replaced our original kinetic scheme with a simpler model that addresses the challenges listed by the reviewer. The new crosslinking mechanism, shown in Figure 2b, includes a burst phase that accommodates the fast crosslinking of much of the EDC-reactive RNC population. Based on the known mechanism of action by EDC, the likely burst-phase events are proposed in Figure 2b (yet not explicitly fit). The slow detectable phase was modeled as two steps, the second of which accounts for the second crosslinking step and is much faster than the first step, hence kinetically inconsequential. We also evaluated the reduced chi-square for the current fits (all values were < 2 , see Suppl. Fig. S2d). The slow step accounts for a hypothetical conformational change necessary to enable covalent crosslinking of the interacting counterparts. Overall, our current simplified model stresses the fact that a fraction of the RNC population crosslinks very fast while another smaller fraction crosslinks slowly. Importantly, the model also acknowledges the experimental observation that some of the RNC population never crosslinks over the timescale of our experiments, as evident in the XL and non-XL gel bands in Figure 2a at the longest crosslinking time of 40 min. The simulations (based on the kinetic parameters derived from the updated curve fitting) are now closely representing the experimental data, as shown in the updated Figure 2d. The close match between experimental data and computer simulations shows that that the overfitting previously identified by the reviewer no longer applies. As a result, kinetic data fitting and computer simulations are now fully consistent with each other. Finally, we kept data interpretation at its simplest and we refrained from over-interpreting the slow steps, as suggested by the reviewer. On the other hand, in the process of updating data analysis, we also noticed that one of the three kinetic experiments was performed under slightly different conditions from the others due to a less effective reaction quenching method. Therefore, to err on the side of safety, we decided to omit this experiment from the presented data set. The data and revised kinetic model are described in the Results section of the manuscript (pages 8 - 9) and illustrated in Figure 2 and supplementary Figure S2.

Comment 2: While the crosslinks of NC to L23 and L29 is convincing (Figure 3), the crosslinks to L24 is much less clear due to high background in the lane. This data needs to be repeated.

Response: We thank the reviewer for this suggestion. We agree with the comments on the high background level of the data in our original Figure 3, pertaining to the interactions between nascent chain and L24 ribosomal protein. To address the comments by the reviewer, we went ahead and carried out several new experiments. Specifically, to minimize the presence of a strong background in the Western blots against ribosomal protein L24, we switched from the previously employed chemiluminescent detection to a more sensitive approach employing fluorescently labeled secondary antibodies. Band detection is now clear and unequivocal for the Western blots of all species analyzed in this work. As a result of the new data collection, we were able to ascertain that full-length PIR (PIR₉₁) interacts only with ribosomal proteins L23 and, weakly, with L29, but not with L24. The new experiments were clearly well worth it and, again, we are very thankful to the reviewer for the suggestion to perform more experiments.

In addition, we also repeated all the Western-blot experiments on the incomplete-chain RNCs, i.e., PIR₅₀ and PIR₇₀. Only interactions with L23, but not with L24 or L29, were detected in the incomplete-chain ribosome-nascent-chain complexes (RNCs). In all, a simple model is emerging according to which the longer nascent protein PIR₉₁ reaches further out within the ribosome, expanding its interactome to L29 while the shorter chains only interact with L23. The new data are discussed in the Results section (pages 10 - 12) and explicitly shown in Figures 3 and 4c, as well as supporting Figure S3.

Comment 3 (part one): The authors constructed a series of charge-reversed mutations on the PIR nascent chain and, in most cases, did not observe major changes in crosslinking efficiency. It wasn't until residues 23-25 were mutated that the authors observed a reduction in crosslinking efficiency. The authors concluded from these data that Coulombic interactions are involved in NC interactions. However, alternative interpretations of the data are not considered.

Response: We took care of these thoughtful comments as follows. In the revised version of the manuscript, we are now clarifying that both Coulombic and non-Coulombic effects may contribute to the disrupted interactions upon converting PIR-1.0.3 into PIR-1.2.3 (page 15, lines 17 - 23). Specifically, the results of Figure 6c suggest that the mutated positively charged side chains of K23 and/or R25 may disrupt *direct* Coulombic interactions with ribosomal proteins. Alternatively, these mutations may cause disruption of interactions via indirect electrostatic effects. For instance, an increased PIR-1.2.3 chain dynamics caused by a high level of intra-chain repulsion in this mutant (bearing 12 negatively charged residues) may lead to an increased chain entropy and result into the disruption of interactions of any kind anywhere in the PIR-1.2.3 chain.

Comment 3 (part two): [...] the correction of crosslinking efficiency by the number of E/D residues is not justified by data, as the author later showed that the position of residues in the nascent chain is also important for crosslinking efficiency.

Response: Indeed, not all EDC-reactive residues exhibit the same fraction of crosslinking, as shown in the plot of Figure 5b (righthandside). The theoretical framework outlined in the manuscript in the section titled “Nascent chains interact with ribosomal proteins” explains the

origin of this effect. Briefly, the fraction of crosslinking is primarily governed by the three independent phenomena: the *intrinsic reactivity* of EDC-reactive functional groups (that depends on the nature of the functional group and on amino-acid sequence), the functional-group solvent exposure (that depends on the specific site and on the system's geometry), and the presence of nascent-protein/ribosomal-protein interactions. These effects are recapitulated in the manuscript equation 1. Now, the literature shows that the intrinsic reactivity of EDC-reactive functional groups is roughly constant under any given solution conditions at constant temperature and pressure (Hoare D.G., Koshland D.E. *J Biol. Chem.* **242**, 2447-2453 (1967)). Therefore, we believe that it is most useful to preserve the plots representing the fraction of crosslinking per EDC-reactive residue (e.g., see Fig. 5b, plot on the righthandside) as in the original version of the manuscript. In this way, the contribution of the remaining two effects, namely solvent exposure and presence of RNC-ribosomal protein interactions, can be most properly and conveniently assessed. This representation the data enables learning about nascent-chain behavior. For instance, we were able to determine that the higher fraction of crosslinking of EDC-reactive residues at longer nascent chains (Fig. 5b) is due to a combination of more solvent exposure and a greater extent of interactions.

Comment 3 (part three): [...] To establish the importance of electrostatic interactions, dependence of the crosslink on ionic strength will be helpful.

Response: This is a helpful suggestion. To further understand the chemical nature of the interactions, we carried out additional experiments at variable ionic strength. Specifically, we assessed the effect of KCl on the interactions involving PIR₅₀, PIR₇₀, full-length PIR (PIR₉₁0.0.0) and the highly negatively-charged variant of full-length PIR (PIR₉₁1.2.3). We found that in all cases the interactions are insensitive to the presence of added KCl, up to 500mM KCl. This result is consistent with ionic-strength-dependent studies (as a function of [NaCl]) carried out in the Cavagnero laboratory a few years ago on PIR RNCs (Knight *et al. ACS Chem. Biol.*, 2013). Given the thick layer of counterions that are known to be present around RNA and RNA-protein complexes (see references from D. Draper's group in the manuscript), this result highlights the fact that there must be a high level of charge shielding close to the ribosomal surface, even at low KCl concentration. This result explains the relatively flat charge-dependent profile of the plot in Figure 6c. The new experiments are now documented and explained in the manuscript (Fig. 7, and page 14 line 23 to page 15 line 7).

In addition, we carried out new investigations on the potential role of Mg⁺². Surprisingly, we found that the extent of RNC-ribosomal-protein interactions decreases significantly at lower total Mg²⁺ concentration and is affected by the net charge of the chain. We propose that Mg²⁺ ions mediate interactions between negatively charged side chains of RNCs and the ribosomal surface, which has a negatively charged surface potential. This exciting result explains why even highly negatively charged RNCs encoding intrinsically disordered protein sequences (e.g., PIR₉₁1.2.3) are able to significantly interact with the ribosome. Notably, as shown in Figure7b, interactions with the ribosomal surface are found to be highly significant across the physiologically relevant Mg⁺² concentration range of 0.8 to 2.2 mM. These new experiments are now documented and explained in the manuscript (Fig. 7, and page 16 line 22 to page 18 line 2).

Comment 3 (part four): [...] In addition, mutations that reduce charge, rather than increase/reverse charge as well as changing the number of crosslink-reactive residues, need to

be tested. Finally, to establish that residues 23-25 make specific interactions with a basic site on the ribosome surface, these residues need to be mutated in the context of WT PIR instead of the hyper-mutated variant.

Response: We agree with all of these excellent suggestions by the reviewer. On the other hand, in this work we are primarily interested in testing whether PIR RNCs with a progressively more negative net charge are able to interact with the ribosome despite its highly negative electrostatic surface potential. This interest motivated the choice of the specific constructs and experimental conditions selected for this study. We acknowledge that studying the effect of reduced net charge and reduced number of crosslinking-reactive residues would be really interesting to further explore the role of electrostatic contributions. In addition, mutations of residues 23-25 in the context of wild-type PIR would also help nailing the role of the positive charge of K23 and R25. On the other hand, we prefer to postpone these excellent additional proposed experiments to future studies. Our purpose of this work is to introduce the reader to the fact that traditional electrostatic effects, involving negative-to-positive charge attractions and negative-to-negative charge repulsions, are non-dominant while Mg^{+2} -mediated interactions are dominant, and likely very strong when RNCs bear a large net negative charge. The detailed role of electrostatics, including the role of RNC positive charges and Mg^{+2} ions, is complex and we plan to separately analyze it in further detail in future studies.

Comment 4 (part one): Presentation of the electrostatic surface potential of the ribosome in Figure 6d is confusing. Is the negatively charged surface dominated by ribosomal RNA, or is it contributed by proteins?

Response: The negatively charged surface of the ribosome is mostly attributable to ribosomal RNA but it is also contributed by ribosomal proteins. Ribosomal proteins are known to exhibit charge-segregation and preferentially display negative charges within their solvent-exposed regions, thus reducing electrostatic repulsion within the ribosomal RNA (Fedyukina, Jennaro and Cavagnero *J. Biol. Chem.* 2014). To more explicitly clarify the electrostatic surface-potential contributions by ribosomal proteins and ribosomal RNA, we generated a new panel (panel e) in Figures 6 of the manuscript. This figure shows a superposition of the ribosomal electrostatic surface potential of the ribosome and the rims of the ribosomal proteins close to the ribosomal exit tunnel. In this way, it is easy to assess surface potential contributed by proteins and by RNA. In this figure, we showed the ribosomal surface potential according to two coloring scales: 0-5 and 0-10 $k_B T$, so that it is easier to appreciate the nonpolar surface patches of the ribosomal proteins in the vicinity of the exit tunnel. We also explicitly explained in the manuscript (page 15, lines 8 – 16) that the highly negative electrostatic surface potential of the 50S ribosomal subunit is dominated by the highly negatively-charged and highly surface-populated ribosomal RNA. Finally, we generated Supplementary Figure S9 and Supplementary movie S2 to visually display the charge segregation of ribosomal proteins and their contribution to the surface electrostatic potential of the ribosome.

Comment 4 (part two): Further, the color scaling of APBS in this figure seems narrow (± 3 kT/e) and may artificially accentuate negative charge. The figure gives the impression that the ribosome surface surrounding the exit site is highly acidic and drove the model in Discussion that NC interactions with the ribosome are electrostatic; however, this conclusion runs counter

to the observation that extensively changing the charge density and nature of PIR nascent chain has no effect on crosslinking efficiency.?

Response: To address this comment by the reviewer, we are now showing, in the new panel e of Figure 6, the ribosomal surface potential according to two coloring scales: 0-5 and 0-10 $k_B T$. As mentioned in the previous response, it is now easier to appreciate the nonpolar surface patches of the ribosomal proteins in the vicinity of the exit tunnel. In the new Supplementary Figure S8, we are also representing the electrostatic potential of the *E. coli* 50S ribosomal subunit with an even wider color scaling, ranging from ± 20 to $0.1 \pm k_B T$. Indeed, the influence of electrostatics turns out to be non-trivial, as the reviewer alludes to. The ribosomal electrostatic surface potential is indeed highly negative (in the case of the entire ribosome as well as near the exit tunnel). On the other hand, the expected high counterion charge density near the ribosomal surface is likely to significantly attenuate electrostatic surface effects arising from +/- attractions and -/- repulsions (as well as any +/+ repulsions). This expectation is consistent with the absence of KCl concentration dependence of the RNC-ribosome interactions reported in Figure 7a. The behavior of chains PIR₉₁-1.0.3 and PIR₉₁-1.0.3 (Fig. 6c) suggests that electrostatic contributions can in some cases be sensed by the chain (though these data do not prove that the RNC-ribosome interactions are of electrostatic nature). The Mg^{+2} dependence studies show that this ion mediates a large fraction of the interactions, depending on the chain characteristics and on the total Mg^{+2} concentration. Therefore, we are now concluding that Mg^{+2} mediates a significant fraction of the interactions, likely contributed by charge-transfer effects. We are also stating that other conventional electrostatic effects may be present, yet more studies are needed to shed additional light on this matter. The above concepts are now explained in the manuscript (pages 16 – 20).

Comment 5: Based on the SDS-PAGE running pattern of NC crosslinks, the authors concluded that the bands represent simultaneous crosslinks of the PIR NC to multiple ribosomal proteins. Additional evidence is needed for this conclusion. The migration of crosslinked protein complexes in SDS-PAGE can vary substantially depending on the site of crosslink and conformation of the crosslinked complex, without invoking the presence of additional crosslink partners.

Response: We thank the reviewer for this particularly thoughtful and important suggestion. Indeed, we more closely analyzed the data on PIR₅₀ and PIR₇₀ and we concluded that indeed the introduction of crosslinks can affect the migration pattern of protein complexes, to some extent. Therefore, we decided to remove the entire former Figure 8 and its accompanying analysis (focusing on the binding of multiple ribosomal proteins to RNCs) from our study, to prevent any data misinterpretation. In addition, we clarified in the manuscript that the presence of crosslinks could affect gel-band migration (page 10, lines 11 – 13).

In response to the comments by Reviewer #2:

Comment 1: The manuscript "Interaction of an intrinsically disordered nascent protein with specific regions of the ribosomal surface" presents an interesting analysis on the interactions between ribosome-bound nascent chains of an intrinsically disordered protein and the proteins around the ribosomal exit tunnel region. By a cross-linking based approach, the authors detail a

relevant pattern of interactions within selected ribosomal proteins, but not others, and the nascent peptide chain. Furthermore, they analyze the dependence of these interactions on the peptide length and its surface charge, using an elegant mutagenesis approach.

I think the points and the conclusions raised in the manuscript are original and sound, and I am convinced it presents an important first step in shedding light in the still relatively unknown universe of ribosomal interactions with peptide nascent chains. I also think the authors do a great job in keeping the text effective and pleasant to read. As a main suggestion for the authors I would encourage them to add more "specificity" to the title of their manuscript, for instance mentioning that the "specific regions of the ribosomal surfaces" are indeed concentrated around the exit tunnel.

Response: We are grateful to this reviewer for all the favorable comments and appreciate the suggestion to add more “specificity” to our manuscript title. Accordingly, we modified the title of the manuscript to: “An intrinsically disordered nascent protein interacts with specific regions of the ribosomal surface near the exit tunnel”.

Comment 2: Also I would find a way to mention that since the study focuses only on protein-protein interactions, keeping protein-rRNA interactions for later.

Response: We thank the reviewer for this insightful comment. We have now included a sentences at the end of the Discussion section (page 20, lines 8 – 10) implying that our study focuses in protein-protein interactions and that nascent-protein-RNA interactions will be separately addressed in future studies.

Comment 3: Additional remarks on two figure legends: - In the Fig.1 the legend of (a) and (b) panels are swapped - In the Fig.4 legend the (e) label is repeated two times and (f) is missing.

Response: Many thanks for these helpful corrections, which we were happy to implement.

In response to the comments by Reviewer #3:

Comment 1 (part one): Most of the data are first rate, and except for the suggested mechanism, their interpretation is sound. I agree with their statement, “The discovery of interactions between nascent proteins lacking signal or arrest sequences and the ribosome is a key step in advancing our knowledge on the influence of the ribosome in protein folding.”, but the present work is a modest advance. The contribution sheds light on what can be done, but, because the protein is intrinsically disordered, the data only dimly illuminate the process. The observations are what one would expect for a disordered protein. In terms of identifying where nascent chains interact with the ribosome, as the authors show in their citation of published efforts, the work is more confirmatory than transformative. Consider the statement ‘this work unravels a new role of the ribosome by showing that specific region of the 50S subunit actively engage in interactions with the nascent protein chain, even in the absence of signal or arrest sequences.’ It would be big news if the nascent chain did NOT interact this part of the ribosome.

Response: We are thankful to the reviewer for the comments on the significance and potential impact of our work. These comments stimulated some productive thinking. Indeed, it is important to eventually extend this research to foldable protein sequences so that the impact of the nascent-chain ribosome interactions can be evaluated in the context of cotranslational protein folding and aggregation. We are, in fact, currently writing a separate manuscript focusing on this very topic. This manuscript will be submitted soon and it will document the interaction-characteristics of folded protein sequences. On the other hand, the present manuscript deliberately focuses on a nascent protein chain that is unable to fold (i.e., a nascent IDP). The focus here is not so much on “what do these interactions do for protein folding/aggregation” but, rather on “the fact that the interactions exist” (maybe not so unexpected but important to establish) and, most significantly, on “the fact that the interactions occur with very specific ribosomal proteins and not with all proteins at the tunnel exit”. The latter finding is novel and unexpected. In addition, new data that we collected since the last manuscript submission (now included in this revised version) show that the interactions are largely mediated by Mg⁺² ions. This latter finding is also largely unexpected, and it contributes to the significance of our story.

Comment 1 (part two): My key concern is the proposed mechanism. The concentration-versus-time plots in Figure 2c are not warranted given the uncertainties in the rate constants listed in Figure 2b. In several instances the uncertainties in the rate constants are larger than the rate constants themselves, and the confidence intervals includes zero, and negative rate constants. The mechanistic discussion should be stricken.

Response: We completely agree with this comments on the crosslinking mechanism. To address this topic, we have now largely simplified and streamlined the mechanistic scheme. The current mechanism acknowledges the presence of a burst phase, leading to a significant population of fully crosslinked adducts, and a slower step, which leads to the generation of a less-efficiently-crosslinking population. The new mechanism has several desirable features, including more reasonable error bars and full consistency with the simulations. The new mechanism is presented in the revised Figure 2 and in the revised Results section of the manuscript (pages 8 – 9).

Comment 2: A problem for the general reader is the lack of detail on stalling. The one sentence on page 19 about the method as it is applied here does not even have a citation.

Response: We thank the reviewer for this pertinent comment. The specific experimental conditions have now been further clarified, and references from [Behrmann, Koch *et al.* 1998] and [Ellis *et al.* 2008] were added to the manuscript.

Comment 3: The manuscript ignores important recent work from the Clark and Shakhnovich labs. Walsh IM, Bowman MA, Soto Santarriaga IF, Rodriguez A, Clark PL: Synonymous codon substitutions perturb cotranslational protein folding in vivo and impair cell fitness. *Proc Natl Acad Sci USA* 2020, 117:3528. Jacobs WM, Shakhnovich EI: Evidence of evolutionary selection for cotranslational folding. *Proc Natl Acad Sci USA* 2017, 114:11434-11439, as well as recent relevant work from the Christodoulou lab. Waudby CA, Wlodarski T, Karyadi M-E, Cassaignau AME, Chan SHS, Wentink AS, Schmidt-Engler JM, Camilloni C, Vendruscolo M, Cabrita LD, Christodoulou J: Systematic mapping of free energy landscapes of a growing filamin domain during biosynthesis. *Proc Natl Acad Sci USA* 2018, 115:9744.

Response: We thank the reviewer for suggesting the addition of the above important references. We were glad to add all the requested citations to the manuscript.

Comment 4: The manuscript is longer than it need be. Judicious editing could shorten it by more than 25% Here are several examples of statements that both identify weaknesses in the work and can be deleted

'Future studies in a more closely in vivo context are needed to confirm the above proposal in living cells.'

'Further analysis based on higher-resolution techniques will shed light on these topics in the future.'

'While not all the species shown may actually be present, the scheme provides an overview of what is possible.'

'Future studies exploring other nascent-protein sequences and employing higher-resolution tools will ascertain the generality of the observed interactions, and will further unveil their role in protein biosynthesis and folding.'

'Interactions of PIR RNCs with rRNA will be addressed elsewhere.'

Response: We thank the reviewer for this editorial advice. As recommended, we removed some of the suggested sentences and we streamlined the manuscript text as much as possible.

In conclusion, we thank all reviewers for their suggestions and careful reading of the manuscript. We believe that the attached revised version of this work (COMMSBIO-20-1417-T) renders it suitable for publication in *Communications Biology*. Please do not hesitate to contact me in case you have any additional requests or questions.

Kindest Regards,

Silvia Cavagnero,
Professor of Chemistry and Biochemistry

REVIEWERS' COMMENTS:

Reviewer #1 (Remarks to the Author):

The authors have satisfactorily addressed all my comments. The manuscript is very thorough in data acquisition and analyses. I appreciate the revised interpretation of data by the authors. The new data on the strong Mg ion dependence of the interaction of nascent chain with the ribosome surface is intriguing.

A fair number of typos and grammatical errors remain in the text, which may require another careful proof reading. Otherwise, I have no objections to publication of the manuscript in its current form.

Reviewer #3 (Remarks to the Author):

The speculative elements of the manuscript have been removed. The revised manuscript is leaner and meaner and will have a bigger impact. The data show where and in what time range disordered proteins interact with the ribosome. It will be important to see if regions from structured proteins interact in these areas. According to the authors they are heading in the that direction.

One suggestion. In the abstract and on page 20, the authors state that the ribosome ACTIVELY engages the nascent chain. I don't understand. It interacts, but what makes it an active interaction? Because it is detectable? Actively should be deleted.

Silvia Cavagnero
Professor of Chemistry
Department of Chemistry, Room 5357
1101 University Avenue
Madison, WI 53706, USA

Phone: (608) 262-5430
E-mail: cavagnero@chem.wisc.edu

RE: Submission of newly revised manuscript
COMMSBIO-20-1417-T
to *Communications Biology*

Madison, August 11 2021

Dear manuscript reviewers,

I am writing to submit a newly revised version of manuscript COMMSBIO-20-1417-T titled “An intrinsically disordered nascent protein interacts with specific regions of the ribosomal surface” as an article to the journal *Communications Biology*. All authors have approved the submission of this revised manuscript.

All recommendations by the reviewers have been addressed. All the requested formatting updates were also implemented, according to the requests listed in the document titled Final Revision Instructions. For convenience, the changes introduced to the manuscript in response to the reviewers’ comments are underlined, in this rebuttal letter.

I am thankful for the several positive comments about this work. I and the coauthors were glad to address the requested revisions. Our specific responses to each of the reviewers’ comments are detailed below.

In response to the comments by Reviewer #1:

Comment: The authors have satisfactorily addressed all my comments. The manuscript is very thorough in data acquisition and analyses. I appreciate the revised interpretation of data by the authors. The new data on the strong Mg ion dependence of the interaction of nascent chain with the ribosome surface is intriguing. A fair number of typos and grammatical errors remain in the text, which may require another careful proof reading. Otherwise, I have no objections to publication of the manuscript in its current form.

Response: We are thankful to this reviewer for the favorable comments on the revised version of our manuscript. To address the reviewer’s recommendation, we have now carefully proofread the manuscript draft and have eliminated all the typographical errors that we could identify.

In response to the comments by Reviewer #3:

Comment: The speculative elements of the manuscript have been removed. The revised manuscript is leaner and meaner and will have a bigger impact. The data show where and in what time range disordered proteins interact with the ribosome. It will be important to see if regions from structured proteins interact in these areas. According to the authors they are heading in the that direction. One suggestion. In the abstract and on page 20, the authors state that the ribosome ACTIVELY engages the nascent chain. I don’t understand. It interacts, but what makes it an active interaction? Because it is

detectable? Actively should be deleted.

Response: We are grateful to this reviewer about the favorable comments on this revised version of our manuscript. In addition, to address the reviewer's recommendation, we removed the word "actively" from the Abstract.

I hope that the attached newly revised version of this work (COMMSBIO-20-1417-T) renders it suitable for publication in *Communications Biology*. Please do not hesitate to contact me in case you have any additional requests or questions.

Kindest Regards,

Silvia Cavagnero,
Professor of Chemistry and Biochemistry